# 1-(1-Arylethylpiperidin-4-yl)thymine Analogs as Antimycobacterial TMPK Inhibitors

**DOI:** 10.3390/molecules25122805

**Published:** 2020-06-17

**Authors:** Yanlin Jian, Fabian Hulpia, Martijn D. P. Risseeuw, He Eun Forbes, Guy Caljon, Hélène Munier-Lehmann, Helena I. M. Boshoff, Serge Van Calenbergh

**Affiliations:** 1Laboratory for Medicinal Chemistry (FFW), Ghent University, Ottergemsesteenweg 460, B-9000 Gent, Belgium; yanlin.jian@ugent.be (Y.J.); fabian.hulpia@ugent.be (F.H.); martijn.risseeuw@ugent.be (M.D.P.R.); 2Tuberculosis Research Section, Laboratory of Clinical Immunology and Microbiology, National Institute of Allergy and Infectious Disease, National Institutes of Health, 9000 Rockville Pike, Bethesda, MD 20892, USA; grace.chun@nih.gov (H.E.F.); hboshoff@niaid.nih.gov (H.I.M.B.); 3Laboratory of Microbiology, Parasitology and Hygiene, University of Antwerp, Universiteitsplein 1 (S7), B-2610 Wilrijk, Belgium; Guy.Caljon@uantwerpen.be; 4Unit of Chemistry and Biocatalysis, Department of Structural Biology and Chemistry, Institut Pasteur, CNRS UMR3523, 28 Rue du Dr. Roux, CEDEX 15 75724 Paris, France; helene.munier-lehmann@pasteur.fr

**Keywords:** tuberculosis, *Mycobacterium tuberculosis*, *Mtb*TMPK, 1-(1-arylethylpiperidin-4-yl)thymine

## Abstract

A series of *Mycobacterium tuberculosis* TMPK (*Mtb*TMPK) inhibitors based on a reported compound **3** were synthesized and evaluated for their capacity to inhibit *Mtb*TMPK catalytic activity and the growth of a virulent *M. tuberculosis* strain (H37Rv). Modifications of the scaffold of **3** failed to afford substantial improvements in *Mtb*TMPK inhibitory activity and antimycobacterial activity. Optimization of the substitution pattern of the D ring of **3** resulted in compound **21j** with improved *Mtb*TMPK inhibitory potency (three-fold) and H37Rv growth inhibitory activity (two-fold). Moving the 3-chloro substituent of **21j** to the *para*-position afforded isomer **21h**, which, despite a 10-fold increase in IC_50_-value, displayed promising whole cell activity (minimum inhibitory concentration (MIC) = 12.5 μM).

## 1. Introduction

Tuberculosis (TB) is an airborne infectious disease caused by *Mycobacterium tuberculosis* (*M. tuberculosis*). Still belonging to the top ten causes of death worldwide, TB was responsible for claiming 1.5 million lives in 2018, thereby preceding AIDS [1]. The World Health Organization (WHO) launched the “END TB” strategy in 2014, aiming at reducing the incidence of TB by 90% and the number of deaths from TB by 95% by 2035 compared with 2015 levels [2]. Nevertheless, the progress towards this sustainable development goal is disappointing, and the continuing increase in drug-resistant TB cases makes the situation more challenging [1,3].

Patients with drug-sensitive TB are currently treated with a combination regimen, consisting of a two-month treatment with first-line agents rifampicin, isoniazid, pyrazinamide and ethambutol, followed by a four-month treatment with rifampicin and isoniazid. Although this schedule has decreased TB mortality, these gains are being threatened by the advent of coinfection with HIV/AIDS, poor patient adherence and a deficient health care system. Additionally, the emergence of multidrug-resistant TB (MDR-TB) and extensively drug-resistant TB (XDR-TB) further erodes the ambitions of the WHO program [1]. In the case of MDR/XDR TB, a wide palette of second- and third-line anti-TB drugs are used, e.g., fluoroquinolones, ethionamide, thioacetazone, clarithromycin and clofazimine [4,5]. However, this regimen not only requires more toxic and costly medications; it also requires a longer treatment duration (up to 24 months), resulting in a poor outcome. Moreover, soon after their introduction, resistance has already developed to the newly approved agents bedaquiline [6] and delamanid [7], with the use of pretomanid restricted to a limited and specific population of patients [8]. Thus, novel anti-TB agents are needed to effectively shorten the treatment regime and cure MDR-/XDR-TB.

Thymidylate kinase, a key enzyme for the synthesis of the DNA building block thymidine-5′-triphosphate, is indispensable for bacterial survival [9,10,11]. The availability of co-crystal structures of *Mtb*TMPK [12,13,14,15,16] has aided the discovery of *Mtb*TMPK inhibitors, featuring both nucleoside [14,17,18,19,20] and non-nucleoside [14,15,20,21] structures.

A previous work from our laboratory started from compound **1** (Figure 1), originally reported by AstraZeneca as an inhibitor of TMPK’s of Gram-positive bacteria [22]. After we found that it also potently inhibited *Mtb*TMPK, a SAR investigation demonstrated that it could be converted to the achiral inhibitor **2** [14]. Further modifications led to the identification of 1-(1-arylethylpiperidin-4-yl)thymine analog **3**, which, compared to **1**, displayed a nine-fold lower minimum inhibitory concentration (MIC) value for H37Rv [14].

In this manuscript, we describe our optimization efforts towards finding potent antimycobacterial agents based on **3** by introducing modifications on the A/B/C/D ring (Figure 2) and by altering the linker part. An overview of the synthesized analogs is presented in Figure 2.

## 2. Results and Discussion

### 2.1. Chemistry

The envisioned analogs were synthesized according to a literature-reported procedure [14]. Briefly, the synthesis encompassed the reductive amination of N^3^-benzyloxymethyl (BOM)-protected piperidinyl thymine with the appropriate aldehyde, followed by trifluoroacetic acid (TFA)-mediated BOM deprotection to yield the desired analogs [14]. The required aldehyde intermediates (Scheme 1) were obtained through the reduction of commercially available substituted methyl phenylacetate esters with LiAlH_4_ [23] and subsequent re-oxidation by pyridinium chlorochromate (PCC) [24,25]. When the required methyl esters were not commercial available, substituted phenoxy- or benzyloxyaryl analogs were obtained through either Chan-Lam coupling of hydroxyphenylacetic methyl ester with a boronic acid [26] or alkylation of the hydroxyphenylacetic methyl ester with benzyl bromide under basic conditions [27].

Due to the low yield of **17a**–**17c** in the Chan-Lam coupling reaction, a different synthetic route was applied for the synthesis of aldehydes **18a**–**18c**, which was based on nucleophilic aromatic substitution of the phenol with the appropriate 4-subsituted fluorobenzene [28]. Then, a PCC-mediated oxidation furnished the desired aldehydes, which were used without further purification in the reductive amination step (Scheme 2).

Compounds **21a**–**21q** and **23** were synthesized via reductive amination of N^3^-BOM-protected piperidinyl thymine with the appropriate aldehyde [14], followed by trifluoroacetic acid (TFA)-mediated BOM deprotection (Scheme 2) [29]. Potential dimerization during the TFA-mediated deprotection [14] was precluded by adding Et_3_SiH as a cation scavenger [29].

Amide **26** was obtained via N-(3-dimethylaminopropyl)-N′-ethylcarbodiimide (EDC)-mediated coupling of **28** with 2-(3-phenoxyphenyl)acetic acid, which was obtained by hydrolysis of the corresponding methyl ester. The BOM group in the resulting amide intermediate was removed by catalytic hydrogenation with Pd/C [30].

A one-step reductive amination of 4-phenylpiperidine with aldehyde **12b** afforded the phenyl analog **27**.

### 2.2. Biological Activity

All synthesized compounds were evaluated for their capacity to inhibit *Mtb*TMPK. Our previously reported analog **3**, the starting point, was resynthesized and its reported inhibitory potency confirmed (Table 1, IC_50_ values of 11 and 17 µM, respectively) [14]. Replacement of the thymine ring by a phenyl (**27**) led to a complete loss of the inhibitory potency. Additionally, repositioning of the thymine ring from the *para* to the *meta*-position of the piperidine ring [20,22] (racemic compound **23**) resulted in a substantial (>10-fold) loss in the inhibitory potency. In-line with earlier observations, with a one-carbon linker [14,21], amide analog **26** displayed a weak enzyme inhibition.

Having established that 1-(1-arylethylpiperidin-4-yl)thymine is preferable for inhibitory potency, our efforts were then directed toward the exploration of the biphenyl ether tail. Deletion of the terminal phenoxy group (analog **21a**) resulted in a 40-fold decrease in inhibitory potency. Docking studies based on the X-ray co-crystal structure of *Mtb*TMPK with our previously reported 1-(piperidin-4-yl)thymine inhibitor (PDB 5NR7) [14] were performed to rationalize the observed SAR. Docking indicated that the loss of a hydrophobic interaction with Tyr39 accounts for the drop in inhibitory potency (Figure 3A). The addition of a methylene moiety between the terminal phenyl ring and C-phenoxy group (analog **21n**) caused a five-fold decrease in the inhibitory potency. The docking pose of compound **21n** in *Mtb*TMPK (Figure 3B) showed that the elongated benzyloxy phenyl resulted in a weaker hydrophobic interaction with Tyr39 than the phenoxy moiety of **3**. Omitting the oxygen atom between the two phenyl rings led to a complete loss of enzyme inhibitory potency (compound **21b**). Moreover, moving the terminal phenoxy ring of **3** to the *ortho* or *para*-position of ring C (compounds **21c**/**21e**) had a negative effect on the inhibitory activity.

Based on these results, further optimization efforts focused on the substitution pattern of the terminal phenyl ring of **3** and **21n**. As shown in Table 2, most of the substituted analogs exhibited a small decrease in inhibitory activity compared to **3**. Of note, the introduction of sterically demanding electron withdrawing substituents (**21k**/**21l**/**21m**) resulted in a significant drop in *Mtb*TMPK inhibitory potency.

Interestingly, the introduction of a 3-chloro (**21j**) but not a 4-chloro substituent (**21h**) afforded a significant improvement in the inhibitory potency due to an edge-to-face π-stacking interaction between the 3-chlorobenzene ring and Tyr39, as found for compound **3** (Figure 4). Introduction of a 3-chloro substituent in the benzyl analog **21n** also had a beneficial impact on the inhibitory activity.

Finally, all compounds were evaluated for their in vitro antimycobacterial activity (Table 3). Consistent with the observations on the enzyme inhibitory activities, modifications of the scaffold of **3** did not yield analogs with superior antimycobacterial activity (**21a**-**21c**, **21e, 21n, 23** and **26**). Remarkably, analog **27**, in which the thymine moiety is replaced by a phenyl ring, showed potent antimycobacterial activity but lacked selectivity, as evidenced by its equipotent cytotoxicity. The introduction of substituents on the distal phenyl ring of **3** afforded several analogs with improved antimycobacterial activity (**21f**–**21j**). Substitution of the D-ring of the benzyloxy analog **21n** also contributed to the growth inhibitory activity. However, the selectivity vs. MRC-5 fibroblasts was modest.

## 3. Materials and Methods

### 3.1. Enzymatic Assay

The enzymatic assay was performed as previously reported on the recombinant purified *Mtb*TMPK [14,33]. As described by Blondin et al. [34], compounds were evaluated by a serial dilution method using the spectrophotometric assay at fixed concentrations of adenosine triphosphate (ATP) (0.5 mM) and thymidine monophosphate (dTMP; 0.05 mM). The reaction medium includes 50-mM Tris-HCl; pH 7.4; 50-mM KCl; 2-mM MgCl_2_; 0.2-mM nicotinamide adenine dinucleotide (NADH); 1-mM phosphoenol pyruvate and 2 units each of coupling enzymes (lactate dehydrogenase, pyruvate kinase and nucleoside diphosphate kinase). From the experimental data, the IC_50_ value was calculated using KaleidaGraph 4.5.3.

### 3.2. Computational Studies

For the molecular modeling, X-ray structure of the *Mtb*TMPK (PDB entry 5NR7 [14]) was analyzed using AutoDock vina and AutodockTools-1.5.6. [35]. In ChemDraw 3D 16.0, the PDB files of all ligands were generated after the energy was minimized (minimum RMS gradient: 0.001). The PDBQT file of the ligands and receptors were prepared by AutodockTools-1.5.6, including atom types, atomic partial charges and the information on the ligand torsional degrees. Using a grid spacing of 0.375 and 60 × 60 × 60 numbers of grid points, the prepared PDBQT files of ligands and receptors were docked (centered on the *Mtb*TMPK active site PHE70 CE2, the coordinates x, y and z were −0.997, 26.240 and −4.528, correspondingly) through the Lamarckian 4.2 method. Each ligand was docked in Autodock vina 3 times, with each time generating 20 possible conformations. Chimera in combination with LigPlus were used to analyze the results.

### 3.3. In Vitro Antituberculosis Assay

The MIC values of all compounds were determined as previously described [36]. In brief, *M. tuberculosis* H37Rv (ATCC 27294) was grown to optical density at 650 nm wavelength (OD_650nm_) 0.2 in Middlebrook 7H9 medium supplemented with 0.4% glucose, 0.03% Bacto casitone and 0.05% Tyloxapol (7H9/glucose/casitone/Tyloxapol) prior to further 1000-fold dilution in fresh medium. Drugs were 2-fold serially diluted in duplicate in 7H9/glucose/casitone/Tyloxapol (50 µL/well) in a concentration range spanning 100-0.049 µM in sterile 96-well U-bottom clear polystyrene microtiter plates. Isoniazid and DMSO as positive and negative controls, respectively. An equal volume (50 µL) of diluted cells was added to the plates with the serial drug dilution. Plates were sealed in Ziplock bags and incubated at 37 °C. After 7–14 days, plates were read with an enlarging inverted mirror plate reader. The MIC was recorded as the concentration that fully inhibited all visible growth.

### 3.4. In Vitro Cytotoxicity Assay

The cytotoxicity of compounds on MRC-5 fibroblasts was performed exactly as previously reported [14].

### 3.5. Chemistry

All reagents and solvents were purchased from standard commercial sources and were of analytical grade. All synthetic compounds described in this study were checked with analytical TLC (Macherey−Nagel precoated F254 aluminum plates, Düren, Germany), visualized under UV light at 254 nm and purified by column chromatography (CC) on a Reveleris X2 (Grace, BÜCHI, Flawil, Switzerland) automated flash unit. All final compounds and some intermediates were measured with Varian Mercury 300/75 MHz (Palo Alto, CA, USA) or a Bruker AVANCE (Fällanden, Zürich, Switzerland) Neo^®^ 400/100 MHz spectrometer at 298.15 K using tetramethylsilane (TMS) as an internal standard. The analysis and confirmation of the final compounds were conducted with ^1^H, ^13^C, HSQC and HMBC NMR spectral data (Appendix A). High-resolution mass spectrometry was performed on a Waters LCT Premier XE^TM^ (Waters, Zellik, Belgium) time-of-flight (TOF) mass spectrometer equipped with a standard electrospray ionization (ESI) and modular LockSpray^TM^ interface (Waters, Zellik, Belgium). The purity of the tested compounds was determined by LC-MS analysis using a Waters AutoPurification system equipped with a Waters Cortecs C18 column (2.7 μm, 100 × 4.6 mm), as was a gradient system of formic acid in H_2_O (0.2%, *v/v*)/MeCN with a gradient of 95:5 to 0:100 in 6.5 min at a flow rate of 1.44 mL/min.

**General procedure A**: Synthesis of biphenyl ether aldehyde building blocks**.** According to a literature report [26], hydroxyphenylacetic ester derivatives (1.0 eq), phenylboronic acid derivatives (3.0 eq.), Cu(OAc)_2_ (2.0 eq.), 4Å molecular sieves (0.18 g/mmol ester) and pyridine (3.0 eq.) in 1,2-dichloroethane (6.0 mL/mmol ester) afforded the biphenyletheracetic ester intermediates. To a solution of the biphenyletheracetic ester intermediates (1.0 eq.) in dry tetrahydrofuran (THF) (6.0 mL/mmol ester intermediate) was added LiAlH_4_ (2.0 eq.) at 0 °C under N_2_ atmosphere, and the resulting mixture was then stirred at room temperature for 1 h [23]. After complete consumption of the starting material, the reaction mixture was quenched with aq. Na^+^/K^+^ tartrate solution (5.0 mL/mmol LiAlH_4_), and the mixture was stirred at room temperature overnight and then filtered. The collected filtrate was dried and concentrated to afford a crude alcohol intermediate, which was oxidized by PCC (2.0 eq.) for 2 h in dichloromethane (DCM) (5.0 mL/mmol PCC) [24,25]. The reaction mixture was filtered through a short silica column. The collected filtrate was concentrated in vacuo and used in the next step without any additional purification.

**General procedure B**: Synthesis of final compounds**.** To a solution of aldehyde intermediates (1.0 eq.) and **28** (1.0 eq.) in 1,2-dichloroethane (33 mL/mmol aldehyde) was added sodium triacetoxyborohydride (2.0 eq.). The resulting mixture was stirred at room temperature for overnight to afford the BOM-protected intermediate [14], which was deprotected with TFA (17 mL/mmol aldehyde) in the presence of triethylsilane [29] (17 mL/mmol aldehyde) at 73 °C for 4 h. After cooling down to room temperature, the reaction mixture was evaporated in vacuo to remove TFA, and the residue was adjusted to pH 6 with 1N aq. HCl. Purification of the resulting mixture by column chromatography gave the desired compounds.

*2-([1,1’-Biphenyl]-4-yl)acetaldehyde* (**6**). To a solution of methyl 2-([1,1’-biphenyl]-4-yl)acetate (0.3 g, 1.3 mmol) in dry THF (7.8 mL) was added LiAlH_4_ (0.10 g, 2.7 mmol) to give alcohol intermediate, which was oxidized with PCC (0.56 g, 2.6 mmol) in DCM (13.0 mL) to yield aldehyde **6** (C_14_H_12_O, 0.22 g, 1.1 mmol).

*2-(2-Phenoxyphenyl)acetaldehyde* (**12a**). Following the general procedure A, methyl 2-(2-hydroxyphenyl)acetate (1.1 g, 8.1 mmol), phenylboronic acid (2.9 g, 24 mmol), Cu(OAc)_2_ (2.9 g, 16 mmol), 4Å molecular sieves (1.5 g) and pyridine (1.9 mL, 24 mmol) in 1,2-dichloroethane (49 mL) afforded the ester intermediate methyl 2-(2-phenoxyphenyl)acetate **10a** (eluent system: 10% ethylacetate in petroleum ether, C_15_H_14_O_3_, 0.40 g, 1.6 mmol, 21% yield). ^1^H NMR (300 MHz, CDCl_3_) *δ* ppm 3.63 (s, 3 H, OCH_3_), 3.72 (s, 2 H, CH_2_), 6.90 (dd, *J* = 8.1, 1.0 Hz, 1 H, Ph), 6.95–7.01 (m, 2 H, Ph), 7.06–7.15 (m, 2 H, Ph), 7.21–7.37 (m, 4 H, Ph). ^13^C NMR (75 MHz, CDCl_3_) *δ* ppm 35.6 (1 C, CH_2_), 51.8 (1 C, OCH_3_), 118.3 (2 C, Ph), 118.8 (1 C, Ph), 123.0 (1 C, Ph), 123.6 (1 C, Ph), 125.8 (1 C, Ph), 128.6 (1 C, Ph), 129.6 (2 C, Ph), 131.4 (1 C, Ph), 155.0 (1 C, Ph), 157.2 (1 C, Ph), 171.7 (1 C, CO). Then, **10a** (0.20 g, 0.83 mmol) was treated with LiAlH_4_ (63 mg, 1.7 mmol) in dry THF (5.0 mL) to give alcohol intermediate, which was oxidized with PCC (0.34 g, 1.6 mmol) in DCM (8.0 mL) to yield aldehyde **12a** (C_14_H_12_O_2_, 0.15 g, 0.70 mmol).

*2-(3-Phenoxyphenyl)acetaldehyde* (**12b**). Following the general procedure A, methyl 2-(3-hydroxyphenyl)acetate (1.1 g, 8.1 mmol), phenylboronic acid (2.9 g, 24 mmol), Cu(OAc)_2_ (2.9 g, 16 mmol), 4Å molecular sieves (1.5 g) and pyridine (1.9 mL, 24 mmol) in 1,2-dichloroethane (49 mL) afforded the ester intermediate methyl 2-(3-phenoxyphenyl)acetate **10b** (eluent system: 10% ethylacetate in petroleum ether, C_15_H_14_O_3_, 0.80 g, 3.3 mmol, 41% yield). ^1^H NMR (300 MHz, CDCl_3_) *δ* ppm 3.62 (s, 2 H, CH_2_), 3.71 (s, 3 H, OCH_3_), 6.90 - 6.95 (m, 1 H, Ph), 6.98 (t, *J* = 2.1 Hz, 1 H, Ph), 7.01–7.07 (m, 3 H, Ph), 7.13 (tt, *J* = 7.4, 1.1 Hz, 1 H, Ph), 7.26–7.40 (m, 3 H, Ph). ^13^C NMR (75 MHz, CDCl_3_) *δ* ppm 40.9 (1 C, CH_2_), 52.0 (1 C, OCH_3_), 117.3 (1 C, Ph), 119.0 (2 C, Ph), 119.7 (1 C, Ph), 123.3 (1 C, Ph), 124.0 (1 C, Ph), 129.7 (3 C, Ph), 135.7 (1 C, Ph), 156.9 (1 C, Ph), 157.4 (1 C, Ph), 171.6 (1 C, CO). Then, **10b** (0.20 g, 0.83 mmol) was treated with LiAlH_4_ (63 mg, 1.7 mmol) in dry THF (5.0 mL) to give alcohol intermediate, which was oxidized with PCC (0.35 g, 1.6 mmol) in DCM (8.0 mL) to yield aldehyde **12b** (C_14_H_12_O_2_, 0.15 g, 0.71 mmol).

*2-(4-Phenoxyphenyl)acetaldehyde* (**12c**). Following the general procedure A, methyl 2-(4-hydroxyphenyl)acetate (0.60 g, 4.4 mmol), phenylboronic acid (1.6 g, 13 mmol), Cu(OAc)_2_ (1.6 g, 8.8 mmol), 4Å molecular sieves (0.70 g) and pyridine (1.1 mL, 13 mmol) in 1,2-dichloroethane (26 mL) afforded the ester intermediate methyl 2-(4-phenoxyphenyl)acetate **10c** (eluent system: 10% ethylacetate in petroleum ether, C_15_H_14_O_3_, 0.66 g, 2.7 mmol, 62% yield). ^1^H NMR (300 MHz, CDCl_3_) *δ* ppm 3.63 (s, 2 H, CH_2_), 3.72 (s, 3 H, OCH_3_), 6.95–7.06 (m, 4 H, Ph), 7.08–7.15 (m, 1 H, Ph), 7.23–7.29 (m, 2 H, Ph), 7.31–7.39 (m, 2 H, Ph). ^13^C NMR (75 MHz, CDCl_3_) *δ* ppm 40.3 (1 C, CH_2_), 52.0 (1 C, OCH_3_), 118.8 (4 C, Ph), 123.2 (1 C, Ph), 128.7 (1 C, Ph), 129.6 (2 C, Ph), 130.5 (2 C, Ph), 156.3 (1 C, Ph), 157.0 (1 C, Ph), 172.0 (1 C, CO). Then, **10c** (0.20 g, 0.83 mmol) was treated with LiAlH_4_ (63 mg, 1.6 mmol) in dry THF (5.0 mL) to give alcohol intermediate, which was oxidized with PCC (0.3 g, 1.4 mmol) in DCM (7.0 mL) to yield aldehyde **12c** (C_14_H_12_O_2_, 0.13 g, 0.62 mmol).

*2-(3-(p-Tolyloxy)phenyl)acetaldehyde* (**12d**). Following the general procedure A, methyl 2-(3-hydroxyphenyl)acetate (0.60g, 4.4 mmol), *p*-tolylboronic acid (1.8 g, 13 mmol), Cu(OAc)_2_ (1.6 g, 8.8 mmol), 4Å molecular sieves (0.79 g) and pyridine (1.1 mL, 13 mmol) in 1,2-dichloroethane (26 mL) afforded the ester intermediate methyl 2-(3-(*p*-tolyloxy)phenyl)acetate **10d** (eluent system: 10% ethylacetate in petroleum ether, C_16_H_16_O_3_, 0.34 g, 1.3 mmol, 30% yield). ^1^H NMR (300 MHz, CDCl_3_) *δ* ppm 2.37 (s, 3 H, CH_3_), 3.62 (s, 2 H, CH_2_), 3.72 (s, 3 H, OCH_3_), 6.89–6.99 (m, 4 H, Ph), 7.00–7.05 (m, 1 H, Ph), 7.14–7.21 (m, 2 H, Ph), 7.25–7.32 (m, 1 H, Ph). ^13^C NMR (75 MHz, CDCl_3_) *δ* ppm 20.6 (1 C, CH_3_), 40.9 (1 C, CH_2_), 51.9 (1 C, OCH_3_), 116.7 (1 C, Ph), 119.1 (3 C, Ph), 123.5 (1 C, Ph), 129.6 (1 C, Ph), 130.1 (2 C, Ph), 132.9 (1 C, Ph), 135.6 (1 C, Ph), 154.4 (1 C, Ph), 157.9 (1 C, Ph), 171.6 (1 C, CO). Then, **10d** (0.20 g, 0.78 mmol) was treated with LiAlH_4_ (59 mg, 1.6 mmol) in dry THF (4.7 mL) to give alcohol intermediate, which was oxidized with PCC (0.28 g, 1.3 mmol) in DCM (6.5 mL) to yield aldehyde **12d** (C_15_H_14_O_2_, 0.13 g, 0.58 mmol).

*2-(3-(3,4-Dichlorophenoxy)phenyl)acetaldehyde* (**12e**). Following the general procedure A, methyl 2-(3-hydroxyphenyl)acetate (0.60 g, 4.4 mmol), (3,4-dichlorophenyl)boronic acid (2.5 g, 13 mmol), Cu(OAc)_2_ (1.6 g, 8.8 mmol), 4Å molecular sieves (0.79 g) and pyridine (1.1 mL, 13 mmol) in 1,2-dichloroethane (26 mL) afforded the ester intermediate methyl 2-(3-(3,4-dichlorophenoxy)phenyl)acetate **10e** (eluent system: 10% ethylacetate in petroleum ether, C_15_H_12_Cl_2_O_3_, 0.73 g, 2.3 mmol, 53% yield). ^1^H NMR (300 MHz, CDCl_3_) *δ* ppm 3.63 (s, 2 H, CH_2_), 3.71 (s, 3 H, OCH_3_), 6.86 (dd, *J* = 8.9, 2.8 Hz, 1 H, Ph), 6.90–6.99 (m, 2 H, Ph), 7.06–7.12 (m, 2 H, Ph), 7.31 (d, *J* = 7.9 Hz, 1 H, Ph), 7.37 (d, *J* = 8.8 Hz, 1 H, Ph). ^13^C NMR (75 MHz, CDCl_3_) *δ* ppm 40.8 (1 C, CH_2_), 52.1 (1 C, OCH_3_), 117.8 (1 C, Ph), 118.0 (1 C, Ph), 120.2 (1 C, Ph), 120.3 (1 C, Ph), 125.1 (2 C, Ph), 130.0 (1 C, Ph), 130.9 (1 C, Ph), 133.1 (1 C, Ph), 136.1 (1 C, Ph), 156.1 (1 C, Ph), 156.3 (1 C, Ph), 171.5 (1 C, CO). Then, **10e** (0.20 g, 0.64 mmol) was treated with LiAlH_4_ (49 mg, 1.3 mmol) in dry THF (3.8 mL) to give alcohol intermediate, which was oxidized with PCC (0.23 g, 1.1 mmol) in DCM (5.5 mL) to yield aldehyde **12e** (C_14_H_10_Cl_2_O_2_, 0.10 g, 0.36 mmol).

*2-(3-(4-Chlorophenoxy)phenyl)acetaldehyde* (**12f**). Following the general procedure A, methyl 2-(3-hydroxyphenyl)acetate (0.60 g, 4.4 mmol), (4-chlorophenyl)boronic acid (2.1 g, 13 mmol), Cu(OAc)_2_ (1.6 g, 8.8 mmol), 4Å molecular sieves (0.79 g) and pyridine (1.1 mL, 13 mmol) in 1,2-dichloroethane (26 mL) afforded the ester intermediate methyl 2-(3-(4-chlorophenoxy)phenyl)acetate **10f** (eluent system: 10% ethylacetate in petroleum ether, C_15_H_13_ClO_3_, 0.40 g, 1.4 mmol, 33% yield). ^1^H NMR (300 MHz, CDCl_3_) *δ* ppm 3.62 (s, 2 H, CH_2_), 3.71 (s, 3 H, OCH_3_), 6.87–6.99 (m, 4 H, Ph), 7.05 (d, *J* = 7.3 Hz, 1 H, Ph), 7.25–7.34 (m, 3 H, Ph). ^13^C NMR (75 MHz, CDCl_3_) *δ* ppm 40.9 (1 C, CH_2_), 52.1 (1 C, OCH_3_), 117.4 (1 C, Ph), 119.8 (1 C, Ph), 120.2 (2 C, Ph), 124.5 (1 C, Ph), 128.4 (1 C, Ph), 129.8 (2 C, Ph), 129.9 (1 C, Ph), 136.0 (1 C, Ph), 155.7 (1 C, Ph), 157.1 (1 C, Ph), 171.6 (1 C, CO). Then, **10f** (0.20 g, 0.72 mmol) was treated with LiAlH_4_ (55 mg, 1.5 mmol) in dry THF (4.3 mL) to give alcohol intermediate, which was oxidized with PCC (0.31 g, 1.5 mmol) in DCM (7.5 mL) to yield aldehyde **12f** (C_14_H_11_ClO_2_, 0.13 g, 0.53 mmol).

*2-(3-(4-Methoxyphenoxy)phenyl)acetaldehyde* (**12g**). Following the general procedure A, methyl 2-(3-hydroxyphenyl)acetate (0.60 g, 4.4 mmol), (4-methoxyphenyl)boronic acid (2.0 g, 13 mmol), Cu(OAc)_2_ (1.6 g, 8.8 mmol), 4Å molecular sieves (0.79 g) and pyridine (1.1 mL, 13 mmol) in 1,2-dichloroethane (26 mL) afforded the ester intermediate methyl 2-(3-(4-methoxyphenoxy)phenyl)acetate **10g** (eluent system: 10% ethylacetate in petroleum ether, C_16_H_16_O_4_, 0.33 g, 1.2 mmol, 28% yield). ^1^H NMR (300 MHz, CDCl_3_) *δ* ppm 3.60 (s, 2 H, CH_2_), 3.70 (s, 3 H, OCH_3_), 3.82 (s, 3 H, (Ph)OCH_3_), 6.82–6.87 (m, 1 H, Ph), 6.88–6.94 (m, 3 H, Ph), 6.95–7.03 (m, 3 H, Ph), 7.22–7.29 (m, 1 H, Ph). ^13^C NMR (75 MHz, CDCl_3_) *δ* ppm 41.0 (1 C, CH_2_), 52.0 (1 C, OCH_3_), 55.6 (1 C, (Ph)OCH_3_), 114.8 (2 C, Ph), 116.0 (1 C, Ph), 118.4 (1 C, Ph), 120.9 (2 C, Ph), 123.2 (1 C, Ph), 129.6 (1 C, Ph), 135.6 (1 C, Ph), 149.8 (1 C, Ph), 155.9 (1 C, Ph), 158.6 (1 C, Ph), 171.7 (1 C, CO). Then, **10g** (0.20 g, 0.74 mmol) was treated with LiAlH_4_ (56 mg, 1.5 mmol) in dry THF (4.4 mL) to give alcohol intermediate, which was oxidized with PCC (0.32 g, 1.5 mmol) in DCM (7.5 mL) to yield aldehyde **12g** (C_15_H_14_O_3_, 0.14 g, 0.58 mmol).

*2-(3-(3-Chlorophenoxy)phenyl)acetaldehyde* (**12h**). Following the general procedure A, methyl 2-(3-hydroxyphenyl)acetate (0.40 g, 2.9 mmol), 3-chlorophenylboronic acid (1.4 g, 8.8 mmol), Cu(OAc)_2_ (1.1 g, 5.9 mmol), 4Å molecular sieves (0.50 g) and pyridine (0.71 mL, 8.8 mmol) in 1,2-dichloroethane (17 mL) afforded the ester intermediate methyl 2-(3-(3-chlorophenoxy)phenyl)acetate **10h** (eluent system: 10% ethylacetate in petroleum ether, C_15_H_13_ClO_3_, 0.28 g, 1.0 mmol, 34% yield). ^1^H NMR (300 MHz, CDCl_3_) *δ* ppm 3.63 (s, 2 H, CH_2_), 3.71 (s, 3 H, OCH_3_), 6.88–6.96 (m, 2 H, Ph), 6.98 (t, *J* = 1.9 Hz, 1 H, Ph), 7.00 (t, *J* = 2.2 Hz, 1 H, Ph), 7.04–7.14 (m, 2 H, Ph), 7.17–7.36 (m, 2 H, Ph). ^13^C NMR (75 MHz, CDCl_3_) *δ* ppm 40.9 (1 C, CH_2_), 52.1 (1 C, OCH_3_), 116.8 (1 C, Ph), 117.9 (1 C, Ph), 118.9 (1 C, Ph), 120.2 (1 C, Ph), 123.3 (1 C, Ph), 124.8 (1 C, Ph), 130.0 (1 C, Ph), 130.5 (1 C, Ph), 135.0 (1 C, Ph), 136.0 (1 C, Ph), 156.5 (1 C, Ph), 158.1 (1 C, Ph), 171.5 (1 C, CO). Then, **10h** (0.20 g, 0.72 mmol) was treated with LiAlH_4_ (55 mg, 1.4 mmol) in dry THF (4.3 mL) to give alcohol intermediate, which was oxidized with PCC (0.19 g, 0.87 mmol) in DCM (4.3 mL) to yield aldehyde **12h** (C_14_H_11_ClO_2_, 0.10 g, 0.41 mmol).

*2-(3-(Benzyloxy)phenyl)acetaldehyde* (**15a**). According to a literature procedure [27] with minor changes, methyl 2-(3-hydroxyphenyl)acetate (0.30 g, 2.2 mmol), benzyl bromide (0.26 mL, 2.2 mmol), K_2_CO_3_ (0.61 g, 4.4 mmol), sodium iodide (33 mg, 0.22 mmol) in dimethylformamide (DMF) (10 mL) at room temperature for overnight afforded the ester intermediate methyl 2-(3-(benzyloxy)phenyl)acetate **13a** (eluent system: 10% ethylacetate in petroleum ether, C_16_H_16_O_3_, 0.33 g, 1.3 mmol, 58% yield). ^1^H NMR (300 MHz, CDCl_3_) *δ* ppm 3.65 (s, 2 H, CH_2_), 3.73 (s, 3 H, OCH_3_), 5.10 (s, 2 H, (Ph)CH_2_O), 6.83–7.05 (m, 3 H, Ph), 7.21–7.52 (m, 6 H, Ph). ^13^C NMR (75 MHz, CDCl_3_) *δ* ppm 41.1 (1 C, CH_2_), 52.0 (1 C, OCH_3_), 69.8 (1 C, (Ph)CH_2_O), 113.4 (1 C, Ph), 115.8 (1 C, Ph), 121.8 (1 C, Ph), 127.5 (2 C, Ph), 127.9 (1 C, Ph), 128.5 (2 C, Ph), 129.5 (1 C, Ph), 135.4 (1 C, Ph), 136.9 (1 C, Ph), 158.9 (1 C, Ph), 171.8 (1 C, CO). Then, **13a** (0.20 g, 0.78 mmol) was treated with LiAlH_4_ (59 mg, 1.6 mmol) in dry THF (4.7 mL) to give alcohol intermediate, which was oxidized with PCC (0.34 g, 1.6 mmol) in DCM (8.0 mL) to yield aldehyde **15a** (C_15_H_14_O_2_, 0.14 g, 0.62 mmol).

*2-(3-((2-Chlorobenzyl)oxy)phenyl)acetaldehyde* (**15b**). Following the procedure as described for **15a**, methyl 2-(3-hydroxyphenyl)acetate (0.40 g, 2.9 mmol), 1-(bromomethyl)-2-chlorobenzene (0.60 g, 2.9 mmol), K_2_CO_3_ (0.81 g, 5.9 mmol), sodium iodide (44 mg, 0.29 mmol) in DMF (13 mL) afforded the ester intermediate methyl 2-(3-((2-chlorobenzyl)oxy)phenyl)acetate **13b** (eluent system: 10% ethylacetate in petroleum ether, C_16_H_15_ClO_3_, 0.26 g, 0.89 mmol, 30% yield). ^1^H NMR (300 MHz, CDCl_3_) *δ* ppm 3.63 (s, 2 H, CH_2_), 3.71 (s, 3 H, OCH_3_), 5.17 (s, 2 H, (Ph)CH_2_O), 6.86–7.01 (m, 3 H, Ph), 7.23–7.36 (m, 3 H, Ph), 7.41 (dd, *J* = 7.0, 2.1 Hz, 1 H, Ph), 7.53–7.63 (m, 1 H, Ph). ^13^C NMR (75 MHz, CDCl_3_) *δ* ppm 41.2 (1 C, CH_2_), 52.1 (1 C, OCH_3_), 67.1 (1 C, (Ph)CH_2_O), 113.4 (1 C, Ph), 116.0 (1 C, Ph), 122.1 (1 C, Ph), 126.9 (1 C, Ph), 128.8 (1 C, Ph), 129.0 (1 C, Ph), 129.3 (1 C, Ph), 129.6 (1 C, Ph), 132.6 (1 C, Ph), 134.7 (1 C, Ph), 135.5 (1 C, Ph), 158.7 (1 C, Ph), 171.8 (1 C, CO). Then, **13b** (0.20 g, 0.69 mmol) was treated with LiAlH_4_ (52 mg, 1.4 mmol) in dry THF (4.1 mL) to give alcohol intermediate, which was oxidized with PCC (0.25 g, 1.2 mmol) in DCM (6.0 mL) to yield aldehyde **15b** (C_15_H_13_ClO_2_, 0.14 g, 0.53 mmol).

*2-(3-((3,4-Dichlorobenzyl)oxy)phenyl)acetaldehyde* (**15c**). Following the procedure as described for **15a**, methyl 2-(3-hydroxyphenyl)acetate (0.40 g, 2.9 mmol), 3,4-dichlorobenzyl bromide (0.70 g, 2.9 mmol), K_2_CO_3_ (0.81 g, 5.9 mmol), sodium iodide (44 mg, 0.29 mmol) in DMF (13 mL) afforded the ester intermediate methyl 2-(3-((3,4-dichlorobenzyl)oxy)phenyl)acetate **13c** (eluent system: 10% ethylacetate in petroleum ether, C_16_H_14_Cl_2_O_3_, 0.75 g, 2.3 mmol, 84% yield). ^1^H NMR (300 MHz, CDCl_3_) *δ* ppm 3.62 (s, 2 H, CH_2_), 3.70 (s, 3 H, OCH_3_), 5.01 (s, 2 H, (Ph)CH_2_O), 6.87 (d, *J* = 8.2 Hz, 1 H, Ph), 6.93 (br. s., 2 H, Ph), 7.21–7.30 (m, 2 H, Ph), 7.41–7.48 (m, 1 H, Ph), 7.54 (s, 1 H, Ph). ^13^C NMR (75 MHz, CDCl_3_) *δ* ppm 41.1 (1 C, CH_2_), 52.1 (1 C, OCH_3_), 68.4 (1 C, (Ph)CH_2_O), 113.4 (1 C, Ph), 115.8 (1 C, Ph), 122.3 (1 C, Ph), 126.5 (1 C, Ph), 129.2 (1 C, Ph), 129.7 (1 C, Ph), 130.5 (1 C, Ph), 131.8 (1 C, Ph), 132.6 (1 C, Ph), 135.6 (1 C, Ph), 137.3 (1 C, Ph), 158.4 (1 C, Ph), 171.7 (1 C, CO). Then, **13c** (0.20 g, 0.62 mmol) was treated with LiAlH_4_ (47 mg, 1.2 mmol) in dry THF (3.7 mL) to give alcohol intermediate, which was oxidized with PCC (0.26 g, 1.2 mmol) in DCM (6.0 mL) to yield aldehyde **15c** (C_15_H_12_Cl_2_O_2_, 0.18 g, 0.60 mmol).

*2-(3-((3-Chlorobenzyl)oxy)phenyl)acetaldehyde* (**15d**). Following the procedure as described for **15a**, methyl 2-(3-hydroxyphenyl)acetate (0.40 g, 2.9 mmol), m-chlorobenzyl bromide (0.60 g, 2.9 mmol), K_2_CO_3_ (0.81 g, 5.9 mmol), sodium iodide (44 mg, 0.29 mmol) in DMF (13 mL) afforded the ester intermediate methyl 2-(3-((3-chlorobenzyl)oxy)phenyl)acetate **13d** (eluent system: 10% ethylacetate in petroleum ether, C_16_H_15_ClO_3_, 0.62 g, 2.1 mmol, 73% yield). ^1^H NMR (400 MHz, CDCl_3_) *δ* ppm 3.61 (s, 2 H, CH_2_), 3.70 (s, 3 H, OCH_3_), 5.03 (s, 2 H, (Ph)CH_2_O), 6.84–6.94 (m, 3 H, Ph), 7.23–7.28 (m, 1 H, Ph), 7.29–7.32 (m, 3 H, Ph), 7.35–7.48 (m, 1 H, Ph). ^13^C NMR (101 MHz, CDCl_3_) *δ* ppm 41.2 (1 C, CH_2_), 52.1 (1 C, OCH_3_), 69.1 (1 C, (Ph)CH_2_O), 113.4 (1 C, Ph), 115.8 (1 C, Ph), 122.1 (1 C, Ph), 125.3 (1 C, Ph), 127.4 (1 C, Ph), 128.0 (1 C, Ph), 129.6 (1 C, Ph), 129.8 (1 C, Ph), 134.5 (1 C, Ph), 135.5 (1 C, Ph), 139.0 (1 C, Ph), 158.6 (1 C, Ph), 171.9 (1 C, CO). Then, **13d** (0.20 g, 0.69 mmol) was treated with LiAlH_4_ (52 mg, 1.4 mmol) in dry THF (4.1 mL) to give alcohol intermediate, which was oxidized with PCC (0.26 g, 1.2 mmol) in DCM (6.0 mL) to yield aldehyde **15d** (C_15_H_13_ClO_2_, 0.14 g, 0.52 mmol).

*2-(3-(4-Cyanophenoxy)phenyl)acetaldehyde* (**18a**). To a solution of methyl 2-(3-hydroxyphenyl)acetate (0.35 g, 2.1 mmol) in THF (13 mL) was added LiAlH_4_ (0.16 g, 4.2 mmol) afforded 3-(2-hydroxyethyl)phenol (0.24 g, 1.4 mmol), which was reacted with 4-fluorobenzonitrile (0.21 g, 1.7 mmol), K_2_CO_3_ (0.40 g, 2.9 mmol) in DMF (10 mL) at 90 °C gave 4-(3-(2-hydroxyethyl)phenoxy)benzonitrile [28] **17a** (eluent system: 25% ethylacetate in petroleum ether, C_15_H_13_NO_2_, 0.14 g, 0.59 mmol, 40% yield). ^1^H NMR (300 MHz, CDCl_3_) *δ* ppm 2.89 (t, *J* = 6.6 Hz, 2 H, CH_2_), 3.89 (t, *J* = 6.4 Hz, 2 H, CH_2_(OH)), 6.89–7.06 (m, 4 H, Ph), 7.11 (dd, *J* = 7.6, 0.59 Hz, 1 H, Ph), 7.36 (t, *J* = 7.4 Hz, 1 H, Ph), 7.49–7.69 (m, 2 H, Ph). ^13^C NMR (75 MHz, CDCl_3_) *δ* ppm 38.9 (1 C, CH_2_), 63.3 (1 C, CH_2_(OH)), 105.8 (1 C, Ph), 110.0 (1 C, Ph), 118.0 (2 C, Ph), 118.3 (1 C, CN), 120.9 (1 C, Ph), 125.7 (1 C, Ph), 130.2 (1 C, Ph), 134.1 (2 C, Ph), 141.4 (1 C, Ph), 154.9 (1 C, Ph), 161.5 (1 C, Ph). Then, **17a** was oxidized by PCC (0.25 g, 1.2 mmol) in DCM (6.0 mL) to give aldehyde **18a** (C_15_H_11_NO_2_, 0.11 g, 0.44 mmol).

*2-(3-(4-Nitrophenoxy)phenyl)acetaldehyde* (**18b**). Following the procedure described for **18a**, 3-(2-hydroxyethyl)phenol (0.30 g, 2.2 mmol), 1-fluoro-4-nitrobenzene (0.37 g, 2.6 mmol) and K_2_CO_3_ (0.60 g, 4.3 mmol) in DMF (16 mL) afforded 2-(3-(4-nitrophenoxy)phenyl)ethan-1-ol **17b** (eluent system: 25% ethylacetate in petroleum ether, C_14_H_13_NO_4_, 0.52 g, 2.0 mmol, 92% yield). ^1^H NMR (300 MHz, CDCl_3_) *δ* ppm 2.90 (t, *J* = 6.4 Hz, 2 H, CH_2_), 3.89 (t, *J* = 6.4 Hz, 2 H, CH_2_(OH)), 6.93–7.07 (m, 4 H, Ph), 7.13 (d, *J* = 7.6 Hz, 1 H, Ph), 7.37 (t, *J* = 7.7 Hz, 1 H, Ph), 8.13–8.24 (m, 2 H, Ph). ^13^C NMR (75 MHz, CDCl_3_) *δ* ppm 38.8 (1 C, CH_2_), 63.2 (1 C, CH_2_(OH)), 117.1 (2 C, Ph), 118.4 (1 C, Ph), 121.0 (1 C, Ph), 125.9 (2 C, Ph), 126.0 (1 C, Ph), 130.3 (1 C, Ph), 136.5 (1 C, Ph), 141.5 (1 C, Ph), 154.9 (1 C, Ph), 163.2 (1 C, Ph). Then, **17b** was oxidized by PCC (0.86 g, 4.0 mmol) in DCM (20 mL) to give aldehyde **18b** (C_14_H_11_NO_4_, 0.36 g, 1.4 mmol).

*2-(3-(4-(Trifluoromethyl)phenoxy)phenyl)acetaldehyde* (**18c**). Following the procedure described for **18a**, 3-(2-hydroxyethyl)phenol (0.30 g, 2.2 mmol), 1-fluoro-4-(trifluoromethyl)benzene (0.43 g, 2.6 mmol) and CsCO_3_ (0.85 g, 2.6 mmol) in DMF (16 mL) at 90 °C gave 2-(3-(4-(trifluoromethyl)phenoxy)phenyl)ethan-1-ol **17c** (eluent system: 25% ethylacetate in petroleum ether, C_15_H_13_F_3_O_2_, 0.13 g, 0.46 mmol, 21% yield). ^1^H NMR (300 MHz, CDCl_3_) *δ* ppm 2.87 (t, *J* = 6.6 Hz, 2 H, CH_2_), 3.86 (t, *J* = 6.6 Hz, 2 H, CH_2_(OH)), 6.89–6.98 (m, 2 H, Ph), 7.01–7.12 (m, 3 H, Ph), 7.33 (t, *J* = 6.6 Hz, 1 H, Ph), 7.58 (d, *J* = 8.8 Hz, 2 H, Ph). ^13^C NMR (75 MHz, CDCl_3_) *δ* ppm 38.9 (1 C, CH_2_), 63.3 (1 C, CH_2_(OH)), 117.7 (1 C, Ph), 117.8 (2 C, Ph), 120.4 (2 C, Ph), 124.2 (q, *J* = 271.5 Hz, 1 C, CF_3_), 125.1 (1 C, Ph), 127.0 (2 C, Ph), 130.0 (1 C, Ph), 141.1 (1 C, Ph), 155.9 (1 C, Ph), 160.3 (1 C, Ph). Then, **17c** was oxidized by PCC (0.20 g, 0.92 mmol) in DCM (4.6 mL) to give aldehyde **18c** (C_15_H_11_F_3_O_2_, 0.12 g, 0.43 mmol).

*5-Methyl-1-(1-phenethylpiperidin-4-yl)pyrimidine-2,4(1H,3H)-dione* (**21a**). Following the general procedure B**,** phenylacetaldehyde (36 mg, 0.30 mmol), **28** (0.10 g, 0.30 mmol) and sodium triacetoxyborohydride (0.13 g, 0.61 mmol) in dichloroethane (10 mL) afforded the BOM-protected intermediate, which was deprotected with TFA (5.0 mL) in the presence of triethylsilane (5.0 mL) at 73 °C for 4 h to give the **21a** (eluent system: 5% MeOH in DCM, 53 mg, 0.17 mmol, 56% yield). ^1^H NMR (300 MHz, DMSO-d_6_) *δ* ppm 1.53–1.93 (m, 7 H, 5-CH_3_, piperdyl-3-yl, piperidyl-5-yl), 2.00–2.16 (m, 2 H, piperidyl-2a-yl, piperidyl-6a-yl), 2.57 (d, *J* = 8.2 Hz, 2 H, CH_2_N), 2.66–2.82 (m, 2 H, PhCH_2_), 3.06 (d, *J* = 10.8 Hz, 2 H, piperidyl-2b-yl, piperidyl-6b-yl), 4.16–4.36 (m, 1 H, piperidyl-4-yl), 7.10–7.37 (m, 5 H, Ph), 7.63 (s, 1 H, H-6), 11.19 (s, 1 H, NH). ^13^C NMR (75 MHz, DMSO-d_6_) *δ* ppm 12.0 (1 C, 5-CH_3_), 29.9 (2 C, piperdyl-3-yl, piperidyl-5-yl), 33.0 (1 C, PhCH_2_), 52.4 (2 C, piperidyl-2-yl, piperidyl-6-yl), 52.6 (1 C, piperidyl-4-yl), 59.3 (1 C, CH_2_N), 108.9 (1 C, C-5), 125.8 (1 C, Ph), 128.2 (2 C, Ph), 128.6 (2 C, Ph), 137.7 (1 C, C-6), 140.4 (1 C, Ph), 150.8 (1 C, C-2), 163.7 (1 C, C-4). HRMS (ESI): *m/z* [M + H]^+^ Calcd. for [C_18_H_23_N_3_O_2_ + H]^+^ 314.1863, found 314.1855.

*1-(1-(2-([1,1’-Biphenyl]-4-yl)ethyl)piperidin-4-yl)-5-methylpyrimidine-2,4(1H,3H)-dione* (**21b**). Following the *general procedure B*, **6** (60 mg, 0.30 mmol), **28** (0.10 g, 0.30 mmol) and sodium triacetoxyborohydride (0.13 g, 0.61 mmol) in dichloroethane (10 mL) afforded the BOM-protected intermediate, which was deprotected with TFA (5.0 mL) in the presence of triethylsilane (5.0 mL) at 73 °C for 4 h to give the **21b** (eluent system: 5% MeOH in DCM, 39 mg, 0.10 mmol, 33% yield). ^1^H NMR (400 MHz, DMSO-d_6_) *δ* ppm 1.72–1.93 (m, 5 H, piperdyl-3a-yl, piperidyl-5a-yl, 5-CH_3_), 1.98–2.20 (m, 2 H, piperdyl-3b-yl, piperidyl-5b-yl), 2.89–3.04 (m, 2 H, PhCH_2_), 3.26–3.61 (m, 4 H, piperidyl-2-yl, piperidyl-6-yl), 4.39–4.53 (m, 1 H, piperidyl-4-yl), 7.33–7.40 (m, 3 H, Ph), 7.46 (t, *J* = 7.6 Hz, 2 H, Ph), 7.59–7.71 (m, 5 H, H-6, Ph), 11.30 (s, 1 H, NH), 2 H (CH_2_N) could not be observed. ^13^C NMR (101 MHz, DMSO-d_6_) *δ* ppm 12.1 (1 C, 5-CH_3_), 27.7 (2 C, piperdyl-3-yl, piperidyl-5-yl), 29.9 (1 C, PhCH_2_), 51.4 (3 C, piperidyl-2-yl, piperidyl-6-yl, piperidyl-4-yl), 109.1 (1 C, C-5), 126.6 (2 C, Ph), 126.8 (2 C, Ph), 127.4 (1 C, Ph), 128.9 (2 C, Ph), 129.3 (2 C, Ph), 137.5 (3 C, C-6, Ph), 139.9 (1 C, Ph), 150.8 (1 C, C-2), 163.7 (1 C, C-4), C (CH_2_N) could not be observed. HRMS (ESI): *m/z* [M + H]^+^ Calcd. for [C_24_H_27_N_3_O_2_ + H]^+^ 390.2176, found 390.2186.

*5-Methyl-1-(1-(2-phenoxyphenethyl)piperidin-4-yl)pyrimidine-2,4(1H,3H)-dione* (**21c**). Following the general procedure B, **12a** (64 mg, 0.30 mmol), **28** (0.10 g, 0.30 mmol) and sodium triacetoxyborohydride (0.13 g, 0.61 mmol) in dichloroethane (10 mL) afforded the BOM-protected intermediate, which was deprotected with TFA (5.0 mL) in the presence of triethylsilane (5.0 mL) at 73 °C for 4 h to give the **21c** (eluent system: 5% MeOH in DCM, 40 mg, 0.098 mmol, 32% yield). ^1^H NMR (300 MHz, DMSO-d_6_) *δ* ppm 1.52–1.62 (m, 2 H, piperdyl-3a-yl, piperidyl-5a-yl), 1.68–1.84 (m, 5 H, 5-CH_3_, piperdyl-3b-yl, piperidyl-5b-yl), 1.92–2.04 (m, 2 H, piperidyl-2a-yl, piperidyl-6a-yl), 2.50–2.55 (m, 2 H, CH_2_N), 2.64–2.75 (m, 2 H, PhCH_2_), 2.92 (d, *J* = 11.1 Hz, 2 H, piperidyl-2b-yl, piperidyl-6b-yl), 4.12–4.26 (m, 1 H, piperidyl-4-yl), 6.85–6.92 (m, 3 H, Ph), 7.02–7.15 (m, 2 H, Ph), 7.17–7.26 (m, 1 H, Ph), 7.29–7.38 (m, 3 H, Ph), 7.57 (d, *J* = 1.2 Hz, 1 H, H-6), 11.16 (s, 1 H, NH). ^13^C NMR (75 MHz, DMSO-d_6_) *δ* ppm 12.0 (1 C, 5-CH_3_), 27.3 (1 C, PhCH_2_), 29.9 (2 C, piperdyl-3-yl, piperidyl-5-yl), 52.3 (2 C, piperidyl-2-yl, piperidyl-6-yl), 52.3 (1 C, piperidyl-4-yl), 57.9 (1 C, CH_2_N), 108.9 (1 C, C-5), 117.1 (2 C, Ph), 119.8 (1 C, Ph), 122.6 (1 C, Ph), 124.2 (1 C, Ph), 127.7 (1 C, Ph), 129.9 (2 C, Ph), 131.1 (1 C, Ph), 131.8 (1 C, Ph), 137.6 (1 C, C-6), 150.8 (1 C, C-2), 153.8 (1 C, Ph), 157.6 (1 C, Ph), 163.6 (1 C, C-4). HRMS (ESI): *m/z* [M + H]^+^ Calcd. for [C_24_H_27_N_3_O_3_ + H]^+^ 406.2125, found 406.2133.

*5-Methyl-1-(1-(4-phenoxyphenethyl)piperidin-4-yl)pyrimidine-2,4(1H,3H)-dione* (**21e**). Following the *general procedure B*, **12c** (64 mg, 0.30 mmol), **28** (0.10 g, 0.30 mmol) and sodium triacetoxyborohydride (0.13 g, 0.61 mmol) in dichloroethane (10 mL) afforded the BOM-protected intermediate, which was deprotected with TFA (5.0 mL) in the presence of triethylsilane (5.0 mL) at 73 °C for 4 h to give the **21e** (eluent system: 5% MeOH in DCM, 65 mg, 0.16 mmol, 53% yield). ^1^H NMR (300 MHz, DMSO-d_6_) *δ* ppm 1.57–1.93 (m, 7 H, 5-CH_3_, piperdyl-3-yl, piperidyl-5-yl), 1.97–2.16 (m, 2 H, piperidyl-2a-yl, piperidyl-6a-yl), 2.54 (d, *J* = 7.3 Hz, 2 H, CH_2_N), 2.64–2.77 (m, 2 H, PhCH_2_), 3.04 (d, *J* = 11.1 Hz, 2 H, piperidyl-2b-yl, piperidyl-6b-yl), 4.19–4.29 (m, 1 H, piperidyl-4-yl), 6.87–6.98 (m, 4 H, Ph), 7.04–7.13 (m, 1 H, Ph), 7.18–7.26 (m, 2 H, Ph), 7.29–7.41 (m, 2 H, Ph), 7.60 (d, *J* = 0.9 Hz, 1 H, H-6), 11.17 (s, 1 H, NH). ^13^C NMR (75 MHz, DMSO-d_6_) *δ* ppm 12.0 (1 C, 5-CH_3_), 29.9 (2 C, piperdyl-3-yl, piperidyl-5-yl), 32.2 (1 C, PhCH_2_), 52.4 (2 C, piperidyl-2-yl, piperidyl-6-yl), 52.6 (1 C, piperidyl-4-yl), 59.3 (1 C, CH_2_N), 108.9 (1 C, C-5), 118.2 (2 C, Ph), 118.7 (2 C, Ph), 123.1 (1 C, Ph), 130.0 (2 C, Ph), 130.1 (2 C, Ph), 135.6 (1 C, Ph), 137.7 (1 C, C-6), 150.8 (1 C, C-2), 154.6 (1 C, Ph), 157.0 (1 C, Ph), 163.7 (1 C, C-4). HRMS (ESI): *m/z* [M + H]^+^ Calcd. for [C_24_H_27_N_3_O_3_ + H]^+^ 406.2125, found 406.2124.

5*-Methyl-1-(1-(3-(p-tolyloxy)phenethyl)piperidin-4-yl)pyrimidine-2,4(1H,3H)-dione* (**21f**). Following the *general procedure B*, **12d** (69 mg, 0.30 mmol), **28** (0.10 g, 0.30 mmol) and sodium triacetoxyborohydride (0.13 g, 0.61 mmol) in dichloroethane (10 mL) afforded the BOM-protected intermediate, which was deprotected with TFA (5.0 mL) in the presence of triethylsilane (5.0 mL) at 73 °C for 4 h to give the **21f** (eluent system: 5% MeOH in DCM, 40 mg, 0.095 mmol, 31% yield). ^1^H NMR (400 MHz, DMSO-d_6_) *δ* ppm 1.64 (d, *J* = 10.6 Hz, 2 H, piperdyl-3a-yl, piperidyl-5a-yl), 1.73 - 1.87 (m, 5 H, 5-CH_3_, piperdyl-3b-yl, piperidyl-5b-yl), 2.05 (t, *J* = 11.1 Hz, 2 H, piperidyl-2a-yl, piperidyl-6a-yl), 2.28 (s, 3 H, PhCH_3_), 2.54 (d, *J* = 8.4 Hz, 2 H, CH_2_N), 2.72 (t, *J* = 7.5 Hz, 2 H, PhCH_2_), 3.02 (d, *J* = 11.5 Hz, 2 H, piperidyl-2b-yl, piperidyl-6b-yl), 4.19–4.29 (m, 1 H, piperidyl-4-yl), 6.77 (d, *J* = 8.1 Hz, 1 H, Ph), 6.87 (s, 1 H, Ph), 6.90 (d, *J* = 8.3 Hz, 2 H, Ph), 6.98 (d, *J* = 7.6 Hz, 1 H, Ph), 7.19 (d, *J* = 8.4 Hz, 2 H, Ph), 7.26 (t, *J* = 7.8 Hz, 1 H, Ph), 7.61 (s, 1 H, H-6), 11.20 (s, 1 H, NH). ^13^C NMR (101 MHz, DMSO-d_6_) *δ* ppm 12.0 (1 C, 5-CH_3_), 20.2 (1 C, PhCH_3_), 30.0 (2 C, piperdyl-3-yl, piperidyl-5-yl), 32.8 (1 C, PhCH_2_), 52.4 (2 C, piperidyl-2-yl, piperidyl-6-yl), 52.5 (1 C, piperidyl-4-yl), 59.0 (1 C, CH_2_N), 108.9 (1 C, C-5), 115.5 (1 C, Ph), 118.4 (1 C, Ph), 118.8 (2 C, Ph), 123.4 (1 C, Ph), 129.7 (1 C, Ph), 130.3 (2 C, Ph), 132.5 (1 C, Ph), 137.7 (1 C, C-6), 142.7 (1 C, Ph), 150.8 (1 C, C-2), 154.2 (1 C, Ph), 157.1 (1 C, Ph), 163.7 (1 C, C-4). HRMS (ESI): *m/z* [M + H]^+^ Calcd. for [C_25_H_29_N_3_O_3_ + H]^+^ 420.2282, found 420.2286.

*1-(1-(3-(3,4-Dichlorophenoxy)phenethyl)piperidin-4-yl)-5-methylpyrimidine-2,4(1H,3H)-dione* (**21g**). Following the general procedure B, **12e** (85 mg, 0.30 mmol), **28** (0.10 g, 0.30 mmol) and sodium triacetoxyborohydride (0.13 g, 0.61 mmol) in dichloroethane (10 mL) afforded the BOM-protected intermediate, which was deprotected with TFA (5.0 mL) in the presence of triethylsilane (5.0 mL) at 73 °C for 4 h to give the **21g** (eluent system: 5% MeOH in DCM, 56 mg, 0.12 mmol, 39% yield). ^1^H NMR (400 MHz, DMSO-d_6_) *δ* ppm 1.79 (s, 3 H, 5-CH_3_), 1.91–2.06 (m, 2 H, piperdyl-3a-yl, piperidyl-5a-yl), 2.10–2.19 (m, 2 H, piperdyl-3b-yl, piperidyl-5b-yl), 2.96–3.08 (m, 2 H, PhCH_2_), 3.09–3.22 (m, 2 H, piperidyl-2a-yl, piperidyl-6a-yl), 3.25–3.37 (m, 2 H, CH_2_N), 3.56–3.73 (m, 2 H, piperidyl-2b-yl, piperidyl-6b-yl), 4.46–4.59 (m, 1 H, piperidyl-4-yl), 7.02 (dd, *J* = 8.9, 2.9 Hz, 2 H, Ph), 7.06 (br. s., 1 H, Ph), 7.15 (d, *J* = 7.3 Hz, 1 H, Ph), 7.30 (d, *J* = 2.8 Hz, 1 H, Ph), 7.34–7.46 (m, 2 H, H-6, Ph), 7.65 (d, *J* = 8.9 Hz, 1 H, Ph), 11.34 (s, 1 H, NH). ^13^C NMR (101 MHz, DMSO-d_6_) *δ* ppm 12.2 (1 C, 5-CH_3_), 27.1 (2 C, piperdyl-3-yl, piperidyl-5-yl), 29.4 (1 C, PhCH_2_), 50.5 (1 C, piperidyl-4-yl), 51.2 (2 C, piperidyl-2-yl, piperidyl-6-yl), 56.3 (1 C, CH_2_N), 109.3 (1 C, C-5), 117.8 (1 C, Ph), 118.7 (1 C, Ph), 119.5 (1 C, Ph), 120.2 (1 C, Ph), 125.16 (d, *J* = 43.5 Hz, 1 C, Ph), 130.6 (1 C, Ph), 131.6 (2 C, Ph), 132.0 (1 C, Ph), 137.4 (1 C, C-6), 139.6 (1 C, Ph), 150.7 (1 C, C-2), 155.8 (1 C, Ph), 156.4 (1 C, Ph), 163.7 (1 C, C-4). HRMS (ESI): *m/z* [M + H]^+^ Calcd. for [C_24_H_25_Cl_2_N_3_O_3_ + H]^+^ 474.1346, found 474.1333.

*1-(1-(3-(4-Chlorophenoxy)phenethyl)piperidin-4-yl)-5-methylpyrimidine-2,4(1H,3H)-dione* (**21h**). Following the general procedure B, **12f** (75 mg, 0.30 mmol), **28** (0.10 g, 0.30 mmol) and sodium triacetoxyborohydride (0.13 g, 0.61 mmol) in dichloroethane (10 mL) afforded the BOM-protected intermediate, which was deprotected with TFA (5.0 mL) in the presence of triethylsilane (5.0 mL) at 73 °C for 4 h to give the **21h** (eluent system: 5% MeOH in DCM, 40 mg, 0.091 mmol, 30% yield). ^1^H NMR (300 MHz, DMSO-d_6_) *δ* ppm 1.57–1.67 (m, 2 H, piperdyl-3a-yl, piperidyl-5a-yl), 1.68–1.87 (m, 5 H, 5-CH_3_, piperdyl-3b-yl, piperidyl-5b-yl), 2.03 (t, *J* = 12.3 Hz, 2 H, piperidyl-2a-yl, piperidyl-6a-yl), 2.54 (d, *J* = 8.5 Hz, 2 H, CH_2_N), 2.72 (t, *J* = 7.4 Hz, 2 H, PhCH_2_), 3.00 (d, *J* = 12.0 Hz, 2 H, piperidyl-2b-yl, piperidyl-6b-yl), 4.22 (tt, *J* = 12.1, 4.5 Hz, 1 H, piperidyl-4-yl), 6.83 (dd, *J* = 7.8, 1.9 Hz, 1 H, Ph), 6.92 (s, 1 H, Ph), 6.96–7.06 (m, 3 H, Ph), 7.24–7.33 (m, 1 H, Ph), 7.38–7.43 (m, 2 H, Ph), 7.59 (s, 1 H, H-6), 11.17 (s, 1 H, NH). ^13^C NMR (75 MHz, DMSO-d_6_) *δ* ppm 12.0 (1 C, 5-CH_3_), 30.0 (2 C, piperdyl-3-yl, piperidyl-5-yl), 32.7 (1 C, PhCH_2_), 52.4 (2 C, piperidyl-2-yl, piperidyl-6-yl), 52.5 (1 C, piperidyl-4-yl), 59.0 (1 C, CH_2_N), 108.9 (1 C, C-5), 116.3 (1 C, Ph), 119.2 (1 C, Ph), 120.0 (2 C, Ph), 124.3 (1 C, Ph), 126.9 (1 C, Ph), 129.8 (2 C, Ph), 129.9 (1 C, Ph), 137.6 (1 C, C-6), 143.0 (1 C, Ph), 150.8 (1 C, C-2), 155.8 (1 C, Ph), 156.1 (1 C, Ph), 163.7 (1 C, C-4). HRMS (ESI): *m/z* [M + H]^+^ Calcd. for [C_24_H_26_ClN_3_O_3_ + H]^+^ 440.1736, found 440.1750.

*1-(1-(3-(4-Methoxyphenoxy)phenethyl)piperidin-4-yl)-5-methylpyrimidine-2,4(1H,3H)-dione* (**21i**). Following the general procedure B, **12g** (74 mg, 0.30 mmol), **28** (0.10 g, 0.30 mmol) and sodium triacetoxyborohydride (0.13 g, 0.61 mmol) in dichloroethane (10 mL) afforded the BOM-protected intermediate, which was deprotected with TFA (5.0 mL) in the presence of triethylsilane (5.0 mL) at 73 °C for 4 h to give the **21i** (eluent system: 5% MeOH in DCM, 40 mg, 0.092 mmol, 30% yield). ^1^H NMR (400 MHz, DMSO-d_6_) *δ* ppm 1.65 (d, *J* = 10.6 Hz, 2 H, piperdyl-3a-yl, piperidyl-5a-yl), 1.74–1.87 (m, 5 H, piperdyl-3b-yl, piperidyl-5b-yl, 5-CH_3_), 2.05 (t, *J* = 11.3 Hz, 2 H, piperidyl-2a-yl, piperidyl-6a-yl), 2.52–2.56 (m, 2 H, CH_2_N), 2.70 (t, *J* = 8.3 Hz, 2 H, PhCH_2_), 3.02 (d, *J* = 11.4 Hz, 2 H, piperidyl-2b-yl, piperidyl-6b-yl), 3.74 (s, 3 H, OCH_3_), 4.19–4.29 (m, 1 H, piperidyl-4-yl), 6.72 (d, *J* = 8.1 Hz, 1 H, Ph), 6.83 (s, 1 H, Ph), 6.92–7.01 (m, 5 H, Ph), 7.24 (t, *J* = 7.9 Hz, 1 H, Ph), 7.61 (s, 1 H, H-6), 11.20 (s, 1 H, NH). ^13^C NMR (101 MHz, DMSO-d_6_) *δ* ppm 12.0 (1 C, 5-CH_3_), 30.0 (2 C, piperdyl-3-yl, piperidyl-5-yl), 32.8 (1 C, PhCH_2_), 52.4 (2 C, piperidyl-2-yl, piperidyl-6-yl), 52.5 (1 C, piperidyl-4-yl), 55.4 (1 C, OCH_3_), 59.0 (1 C, CH_2_N), 108.9 (1 C, C-5), 114.7 (1 C, Ph), 115.0 (2 C, Ph), 117.6 (1 C, Ph), 120.6 (2 C, Ph), 123.0 (1 C, Ph), 129.6 (1 C, Ph), 137.7 (1 C, C-6), 142.6 (1 C, Ph), 149.4 (1 C, Ph), 150.8 (1 C, C-2), 155.5 (1 C, Ph), 157.9 (1 C, Ph), 163.7 (1 C, C-4). HRMS (ESI): *m/z* [M + H]^+^ Calcd. for [C_25_H_29_N_3_O_4_ + H]^+^ 436.2231, found 436.2234.

*1-(1-(3-(3-Chlorophenoxy)phenethyl)piperidin-4-yl)-5-methylpyrimidine-2,4(1H,3H)-dione* (**21j**). Following the general procedure B, **12h** (75 mg, 0.30 mmol), **28** (0.10 g, 0.30 mmol) and sodium triacetoxyborohydride (0.13 g, 0.61 mmol) in dichloroethane (10 mL) afforded the BOM-protected intermediate, which was deprotected with TFA (5.0 mL) in the presence of triethylsilane (5.0 mL) at 73 °C for 4 h to give the **21j** (eluent system: 5% MeOH in DCM, 63 mg, 0.14 mmol, 47% yield). ^1^H NMR (400 MHz, DMSO-d_6_) *δ* ppm 1.65 (d, *J* = 9.8 Hz, 2 H, piperdyl-3a-yl, piperidyl-5a-yl), 1.71–1.89 (m, 5 H, piperdyl-3b-yl, piperidyl-5b-yl, 5-CH_3_), 2.06 (t, *J* = 11.0 Hz, 2 H, piperidyl-2a-yl, piperidyl-6a-yl), 2.53–2.62 (m, 2 H, CH_2_N), 2.69–2.79 (m, 2 H, PhCH_2_), 3.03 (d, *J* = 11.6 Hz, 2 H, piperidyl-2b-yl, piperidyl-6b-yl), 4.17–4.31 (m, 1 H, piperidyl-4-yl), 6.88 (dd, *J* = 8.1, 1.8 Hz, 1 H, Ph), 6.92–7.00 (m, 2 H, Ph), 7.02 (t, *J* = 2.2 Hz, 1 H, Ph), 7.08 (d, *J* = 7.8 Hz, 1 H, Ph), 7.15–7.21 (m, 1 H, Ph), 7.33 (t, *J* = 7.8 Hz, 1 H, Ph), 7.37–7.43 (m, 1 H, Ph), 7.60 (d, *J* = 0.9 Hz, 1 H, H-6), 11.20 (s, 1 H, NH). ^13^C NMR (101 MHz, DMSO-d_6_) *δ* ppm 12.0 (1 C, 5-CH_3_), 30.0 (2 C, piperdyl-3-yl, piperidyl-5-yl), 32.7 (1 C, PhCH_2_), 52.4 (3 C, piperidyl-2-yl, piperidyl-6-yl, piperidyl-4-yl), 58.9 (1 C, CH_2_N), 108.9 (1 C, C-5), 116.7 (1 C, Ph), 116.8 (1 C, Ph), 118.0 (1 C, Ph), 119.6 (1 C, Ph), 123.0 (1 C, Ph), 124.7 (1 C, Ph), 130.0 (1 C, Ph), 131.4 (1 C, Ph), 133.9 (1 C, Ph), 137.6 (1 C, C-6), 143.1 (1 C, Ph), 150.8 (1 C, C-2), 155.5 (1 C, Ph), 158.1 (1 C, Ph), 163.7 (1 C, C-4). HRMS (ESI): *m/z* [M + H]^+^ Calcd. for [C_24_H_26_ClN_3_O_3_ + H]^+^ 440.1736, found 440.1740.

*4-(3-(2-(4-(5-Methyl-2,4-dioxo-3,4-dihydropyrimidin-1(2H)-yl)piperidin-1- yl)ethyl)phenoxy)benzonitrile* (**21k**). Following the general procedure B, **18a** (72 mg, 0.30 mmol), **28** (0.10 g, 0.30 mmol) and sodium triacetoxyborohydride (0.13 g, 0.61 mmol) in dichloroethane (10 mL) afforded the BOM-protected intermediate, which was deprotected with TFA (5.0 mL) in the presence of triethylsilane (5.0 mL) at 73 °C for 4 h to give the **21k** (eluent system: 5% MeOH in DCM, 50 mg, 0.12 mmol, 43% yield). ^1^H NMR (400 MHz, DMSO-d_6_) *δ* ppm 1.79 (s, 3 H, 5-CH_3_), 2.00 (d, *J* = 10.8 Hz, 2 H, piperdyl-3a-yl, piperidyl-5a-yl), 2.21 (d, *J* = 11.6 Hz, 2 H, piperdyl-3b-yl, piperidyl-5b-yl), 3.05 (br. s., 2 H, PhCH_2_), 3.15 (br. s., 2 H, piperidyl-2a-yl, piperidyl-6a-yl), 3.32 (s, 2 H, CH_2_N), 3.66 (br. s., 2 H, piperidyl-2b-yl, piperidyl-6b-yl), 4.44–4.63 (m, 1 H, piperidyl-4-yl), 7.03–7.16 (m, 4 H, Ph), 7.21 (d, *J* = 6.6 Hz, 1 H, Ph), 7.35–7.53 (m, 2 H, Ph, H-6), 7.86 (d, *J* = 8.4 Hz, 2 H, Ph), 11.32 (br. s., 1 H, NH). ^13^C NMR (101 MHz, DMSO-d_6_) *δ* ppm 12.2 (1 C, 5-CH_3_), 27.1 (2 C, piperdyl-3-yl, piperidyl-5-yl), 29.3 (1 C, PhCH_2_), 50.3 (1 C, piperidyl-4-yl), 51.2 (2 C, piperidyl-2-yl, piperidyl-6-yl), 56.2 (1 C, CH_2_N), 105.2 (1 C, Ph), 109.3 (1 C, C-5), 118.1 (2 C, Ph), 118.7 (2 C, Ph, CN), 120.5 (1 C, Ph), 125.6 (1 C, Ph), 130.7 (1 C, Ph), 134.7 (2 C, Ph), 137.3 (1 C, C-6), 139.7 (1 C, Ph), 150.7 (1 C, C-2), 154.7 (1 C, Ph), 161.0 (1 C, Ph), 163.6 (1 C. C-4). HRMS (ESI): *m/z* [M + H]^+^ Calcd. for [C_25_H_26_N_4_O_3_ + H]^+^ 431.2078, found 431.2085.

*5-Methyl-1-(1-(3-(4-nitrophenoxy)phenethyl)piperidin-4-yl)pyrimidine-2,4(1H,3H)-dione* (**21l**). Following the general procedure B, **18b** (78 mg, 0.30 mmol), **28** (0.10 g, 0.30 mmol) and sodium triacetoxyborohydride (0.13 g, 0.61 mmol) in dichloroethane (10 mL) afforded the BOM-protected intermediate, which was deprotected with TFA (5.0 mL) in the presence of triethylsilane (5.0 mL) at 73 °C for 4 h to give the **21l** (eluent system: 5% MeOH in DCM, 67 mg, 0.15 mmol, 40% yield). ^1^H NMR (400 MHz, DMSO-d_6_) *δ* ppm 1.79 (s, 3 H, 5-CH_3_), 1.93–2.06 (m, 2 H, piperdyl-3a-yl, piperidyl-5a-yl), 2.20 (d, *J* = 11.6 Hz, 2 H, piperdyl-3b-yl, piperidyl-5b-yl), 2.98–3.08 (m, 2 H, PhCH_2_), 3.09–3.24 (m, 2 H, piperidyl-2a-yl, piperidyl-6a-yl), 3.41–3.55 (m, 2 H, CH_2_N), 3.67 (d, *J* = 10.6 Hz, 2 H, piperidyl-2b-yl, piperidyl-6b-yl), 4.44–4.62 (m, 1 H, piperidyl-4-yl), 7.09–7.17 (m, 4 H, Ph), 7.24 (d, *J* = 7.5 Hz, 1 H, Ph), 7.38 (s, 1 H, H-6), 7.49 (t, *J* = 7.8 Hz, 1 H, Ph), 8.27 (d, *J* = 9.1 Hz, 2 H, Ph), 11.32 (s, 1 H, NH). ^13^C NMR (101 MHz, DMSO-d_6_) *δ* ppm 12.2 (1 C, 5-CH_3_), 27.0 (2 C, piperdyl-3-yl, piperidyl-5-yl), 29.3 (1 C, PhCH_2_), 50.5 (1 C, piperidyl-4-yl), 51.2 (2 C, piperidyl-2-yl, piperidyl-6-yl), 56.2 (1 C, CH_2_N), 109.3 (1 C, C-5), 117.5 (2 C, Ph), 118.9 (1 C, Ph), 120.7 (1 C, Ph), 125.9 (1 C, Ph), 126.2 (2 C, Ph), 130.8 (1 C, Ph), 137.3 (1 C, C-6), 139.4 (1 C, Ph), 142.3 (1 C, Ph), 150.7 (1 C, C-2), 154.5 (1 C, Ph), 162.7 (1 C, Ph), 163.6 (1 C, C-4). HRMS (ESI): *m/z* [M + H]^+^ Calcd. for [C_24_H_26_N_4_O_5_ + H]^+^ 451.1976, found 451.1971.

*5-Methyl-1-(1-(3-(4-(trifluoromethyl)phenoxy)phenethyl)piperidin-4-yl)pyrimidine-2,4(1H,3H)-dione* (**21m**). Following the *general procedure B*, **18c** (85 mg, 0.30 mmol), **28** (0.10 g, 0.30 mmol) and sodium triacetoxyborohydride (0.13 g, 0.61 mmol) in dichloroethane (10 mL) afforded the BOM-protected intermediate, which was deprotected with TFA (5.0 mL) in the presence of triethylsilane (5.0 mL) at 73 °C for 4 h to give the **21m** (eluent system: 5% MeOH in DCM, 64 mg, 0.14 mmol, 44% yield). ^1^H NMR (400 MHz, DMSO-d_6_) *δ* ppm 1.79 (s, 3 H, 5-CH_3_), 2.00 (d, *J* = 12.0 Hz, 2 H, piperdyl-3a-yl, piperidyl-5a-yl), 2.20 (d, *J* = 11.4 Hz, 2 H, piperdyl-3b-yl, piperidyl-5b-yl), 2.99 - 3.08 (m, 2 H, PhCH_2_), 3.16 (d, *J* = 9.5 Hz, 2 H, piperidyl-2a-yl, piperidyl-6a-yl), 3.39–3.55 (m, 2 H, CH_2_N), 3.67 (d, *J* = 10.6 Hz, 2 H, piperidyl-2b-yl, piperidyl-6b-yl), 4.47–4.59 (m, 1 H, piperidyl-4-yl), 7.05 (d, *J* = 7.9 Hz, 1 H, Ph), 7.10–7.20 (m, 4 H, Ph), 7.39 (br. s., 1 H, H-6), 7.44 (t, *J* = 7.8 Hz, 1 H, Ph), 7.75 (d, *J* = 8.5 Hz, 2 H, Ph), 11.32 (s, 1 H, NH). ^19^F NMR (377 MHz, DMSO-d_6_) *δ* ppm -60.16 (s, 3 F). ^13^C NMR (101 MHz, *DMSO-d_6_*) *δ* ppm 12.2 (1 C, 5-CH_3_), 27.1 (2 C, piperdyl-3-yl, piperidyl-5-yl), 29.3 (1 C, PhCH_2_), 50.4 (1 C, piperidyl-4-yl), 51.1 (2 C, piperidyl-2-yl, piperidyl-6-yl), 56.3 (1 C, CH_2_N), 109.3 (1 C, C-5), 118.0 (2 C, Ph), 118.4 (1 C, Ph), 120.2 (1 C, Ph), 123.20 (q, *J* = 32.6 Hz, 1 C, Ph), 124.27 (q, *J* = 271.3 Hz, 1 C, CF_3_), 125.2 (1 C, Ph), 127.5 (2 C, Ph), 130.6 (1 C, Ph), 137.3 (1 C, C-6), 139.6 (1 C, Ph), 150.7 (1 C, C-2), 155.2 (1 C, Ph), 160.3 (1 C, Ph), 163.6 (1 C, C-4). HRMS (ESI): *m/z* [M + H]^+^ Calcd. for [C_25_H_26_F_3_N_3_O_3_ + H]^+^ 474.1999, found 474.1989.

*1-(1-(3-(Benzyloxy)phenethyl)piperidin-4-yl)-5-methylpyridine-2,4(1H,3H)-dione* (**21n**). Following the general procedure B, **15a** (69 mg, 0.30 mmol), **28** (0.10 g, 0.30 mmol) and sodium triacetoxyborohydride (0.13 g, 0.61 mmol) in dichloroethane (10 mL) afforded the BOM-protected intermediate, which was deprotected with TFA (5.0 mL) in the presence of triethylsilane (5.0 mL) at 73 °C for 4 h to give the **21n** (eluent system: 5% MeOH in DCM, 58 mg, 0.14 mmol, 46% yield). ^1^H NMR (400 MHz, DMSO-d_6_) *δ* ppm 1.66 (d, *J* = 10.8 Hz, 2 H, piperdyl-3a-yl, piperidyl-5a-yl), 1.74–1.90 (m, 5 H, 5-CH_3_, piperdyl-3b-yl, piperidyl-5b-yl), 2.07 (t, *J* = 11.3 Hz, 2 H, piperidyl-2a-yl, piperidyl-6a-yl), 2.55 (t, *J* = 9.0 Hz, 2 H, CH_2_N), 2.70 (t, *J* = 7.4 Hz, 2 H, PhCH_2_), 3.04 (d, *J* = 11.3 Hz, 2 H, piperidyl-2b-yl, piperidyl-6b-yl), 4.20–4.31 (m, 1 H, piperidyl-4-yl), 5.08 (s, 2 H, (Ph)CH_2_O), 6.82 (t, *J* = 7.8 Hz, 2 H, Ph), 6.90 (s, 1 H, Ph), 7.19 (t, *J* = 7.9 Hz, 1 H, Ph), 7.30–7.35 (m, 1 H, Ph), 7.39 (t, *J* = 7.4 Hz, 2 H, Ph), 7.43–7.48 (m, 2 H, Ph), 7.63 (s, 1 H, H-6), 11.20 (s, 1 H, NH).^13^C NMR (101 MHz, DMSO-d_6_) *δ* ppm 12.0 (1 C, 5-CH_3_), 30.0 (2 C, piperdyl-3-yl, piperidyl-5-yl), 33.0 (1 C, PhCH_2_), 52.4 (2 C, piperidyl-2-yl, piperidyl-6-yl), 52.4 (1 C, piperidyl-4-yl), 59.2 (1 C, CH_2_N), 69.0 (1 C, (Ph)CH_2_O), 108.9 (1 C, C-5), 112.1 (1 C, Ph), 115.2 (1 C, Ph), 121.1 (1 C, Ph), 127.7 (2 C, Ph), 127.8 (1 C, Ph), 128.4 (2 C, Ph), 129.2 (1 C, Ph), 137.2 (1 C, Ph), 137.7 (1 C, C-6), 142.0 (1 C, Ph), 150.8 (1 C, C-2), 158.4 (1 C, Ph), 163.7 (1 C, C-4). HRMS (ESI): *m/z* [M + H]^+^ Calcd. for [C_25_H_29_N_3_O_3_ + H]^+^ 420.2282, found 420.2277.

*1-(1-(3-((2-Chlorobenzyl)oxy)phenethyl)piperidin-4-yl)-5-methylpyrimidine-2,4(1H,3H)-dione* (**21o**). Following the general procedure B, **15b** (79 mg, 0.30 mmol), **28** (0.10 g, 0.30 mmol) and sodium triacetoxyborohydride (0.13 g, 0.61 mmol) in dichloroethane (10 mL) afforded the BOM-protected intermediate, which was deprotected with TFA (5.0 mL) in the presence of triethylsilane (5.0 mL) at 73 °C for 4 h to give the **21o** (eluent system: 5% MeOH in DCM, 90 mg, 0.20 mmol, 65% yield). ^1^H NMR (400 MHz, DMSO-d_6_) *δ* ppm 1.73–2.23 (m, 7 H, piperdyl-3-yl, piperidyl-5-yl, 5-CH_3_), 2.92 (br. s., 2 H, PhCH_2_), 3.21–3.67(m, 4 H, piperidyl-2-yl, piperidyl-6-yl), 4.35–4.68 (m, 1 H, piperidyl-4-yl), 5.12 (s, 2 H, (Ph)CH_2_O), 6.85–6.93 (m, 2 H, Ph), 6.97 (br. s., 1 H, Ph), 7.26 (t, *J* = 7.8 Hz, 1 H, Ph), 7.36–7.49 (m, 4 H, Ph), 7.52 (s, 1 H, H-6), 11.31 (br. s., 1 H, NH), 2 H (CH_2_N) could not be observed. ^13^C NMR (101 MHz, DMSO-d_6_) *δ* ppm 12.2 (1 C, 5-CH_3_), 27.0 (2 C, piperdyl-3-yl, piperidyl-5-yl), 30.2 (1 C, PhCH_2_), 50.4 (1 C, piperidyl-4-yl), 51.5 (2 C, piperidyl-2-yl, piperidyl-6-yl), 68.2 (1 C, (Ph)CH_2_O), 109.2 (1 C, C-5), 112.7 (1 C, Ph), 115.6 (1 C, Ph), 121.4 (1 C, Ph), 126.2 (2 C, Ph), 127.3 (1 C, Ph), 127.8 (1 C, Ph), 129.7 (1 C, Ph), 130.4 (1 C, Ph), 133.1 (1 C, Ph), 137.5 (1 C, C-6), 139.7 (1 C, Ph), 150,8 (1 C, C-2), 158.3 (1 C, Ph), 163.7 (1 C, C-4), C (CH_2_N) could not be observed. HRMS (ESI): *m/z* [M + H]^+^ Calcd. for [C_25_H_28_ClN_3_O_3_ + H]^+^ 454.1892, found 454.1902.

*1-(1-(3-((3,4-Dichlorobenzyl)oxy)phenethyl)piperidin-4-yl)-5-methylpyrimidine-2,4(1H,3H)-dione* (**21p**). Following the general procedure B, **15c** (90 mg, 0.30 mmol), **28** (0.10 g, 0.30 mmol) and sodium triacetoxyborohydride (0.13 g, 0.61 mmol) in dichloroethane (10 mL) afforded the BOM-protected intermediate, which was deprotected with TFA (5.0 mL) in the presence of triethylsilane (5.0 mL) at 73 °C for 4 h to give the **21p** (eluent system: 5% MeOH in DCM, 55 mg, 0.11 mmol, 37% yield). ^1^H NMR (400 MHz, DMSO-d_6_) *δ* ppm 1.79 (s, 3 H, 5-CH_3_), 1.92–2.06 (m, 2 H, piperdyl-3a-yl, piperidyl-5a-yl), 2.08–2.24 (m, 2 H, piperdyl-3b-yl, piperidyl-5b-yl), 2.85–3.02 (m, 2 H, PhCH_2_), 3.06–3.22 (m, 2 H, piperidyl-2a-yl, piperidyl-6a-yl), 3.24–3.34 (m, 2 H, CH_2_N), 3.65 (br. s., 2 H, piperidyl-2b-yl, piperidyl-6b-yl), 4.45–4.59 (m, 1 H, piperidyl-4-yl), 5.13 (s, 2 H, (Ph)CH_2_O), 6.85–6.95 (m, 2 H, Ph), 6.98 (br. s., 1 H, Ph), 7.28 (t, *J* = 7.9 Hz, 1 H, Ph), 7.38–7.48 (m, 2 H, H-6, Ph), 7.68 (d, *J* = 8.3 Hz, 1 H, Ph), 7.73 (d, *J* = 1.5 Hz, 1 H, Ph), 11.33 (s, 1 H, NH). ^13^C NMR (101 MHz, DMSO-d_6_) *δ* ppm 12.2 (1 C, 5-CH_3_), 27.2 (2 C, piperdyl-3-yl, piperidyl-5-yl), 29.7 (1 C, PhCH_2_), 50.6 (1 C, piperidyl-4-yl), 51.2 (2 C, piperidyl-2-yl, piperidyl-6-yl), 56.5 (1 C, CH_2_N), 67.5 (1 C, (Ph)CH_2_O), 109.3 (1 C, C-5), 112.8 (1 C, Ph), 115.5 (1 C, Ph), 121.5 (1 C, Ph), 127.8 (1 C, Ph), 129.4 (1 C, Ph), 129.8 (1 C, Ph), 130.4 (1 C, Ph), 130.7 (1 C, Ph), 131.1 (1 C, Ph), 137.4 (1 C, C-6), 138.3 (2 C, Ph), 150.7 (1 C, C-2), 158.2 (1 C, Ph), 163.7 (1 C, C-4). HRMS (ESI): *m/z* [M + H]^+^ Calcd. for [C_25_H_27_Cl_2_N_3_O_3_ + H]^+^ 488.1502, found 488.1518.

*1-(1-(3-((3-Chlorobenzyl)oxy)phenethyl)piperidin-4-yl)-5-methylpyrimidine-2,4(1H,3H)-dione* (**21q**). Following the general procedure B, **15d** (79 mg, 0.30 mmol), **28** (0.10 g, 0.30 mmol) and sodium triacetoxyborohydride (0.13 g, 0.61 mmol) in dichloroethane (10 mL) afforded the BOM-protected intermediate, which was deprotected with TFA (5.0 mL) in the presence of triethylsilane (5.0 mL) at 73 °C for 4 h to give the **21q** (eluent system: 5% MeOH in DCM, 38 mg, 0.084 mmol, 28% yield). ^1^H NMR (400 MHz, DMSO-d_6_) *δ* ppm 1.78 (s, 5 H, 5-CH_3_, piperdyl-3a-yl, piperidyl-5a-yl), 2.00 (br. s., 2 H, piperdyl-3b-yl, piperidyl-5b-yl), 2.84 (br. s., 2 H, PhCH_2_), 4.31–4.46 (m, 1 H, piperidyl-4-yl), 5.12 (s, 2 H, (Ph)CH_2_O), 6.87 (t, *J* = 8.3 Hz, 2 H, Ph), 6.94 (br.s., 1 H, Ph), 7.23 (t, *J* = 7.8 Hz, 1 H, Ph), 7.37–7.47 (m, 3 H, Ph), 7.50–7.59 (m, 2 H, Ph, H-6), 11.26 (s, 1 H, NH), 2 H (CH_2_N) and 4 H (piperidyl-2-yl, piperidyl-6-yl) could not be observed. ^13^C NMR (101 MHz, DMSO-d_6_) *δ* ppm 12.1 (1 C, 5-CH_3_), 28.6 (2 C, piperdyl-3-yl, piperidyl-5-yl), 31.3 (1 C, PhCH_2_), 51.5 (1 C, piperidyl-4-yl), 51.7 (2 C, piperidyl-2-yl, piperidyl-6-yl), 68.1 (1 C, (Ph)CH_2_O), 109.0 (1 C, C-5), 112.5 (1 C, Ph), 115.4 (1 C, Ph), 121.3 (1 C, Ph), 126.1 (1 C, Ph), 127.3 (1 C, Ph), 127.7 (1 C, Ph), 129.5 (1 C, Ph), 130.4 (1 C, Ph), 133.1 (1 C, Ph), 137.5 (1 C, C-6), 139.7 (1 C, Ph), 150.8 (1 C, C-2), 158.2 (1 C, Ph), 163.7 (1 C, C-4), C (CH_2_N) and C (Ph) could not be observed. HRMS (ESI): *m/z* [M + H]^+^ Calcd. for [C_25_H_28_ClN_3_O_3_ + H]^+^ 454.1892, found 454.1899.

*5-Methyl-1-(1-(3-phenoxyphenethyl)piperidin-3-yl)pyrimidine-2,4(1H,3H)-dione* (**23**). Following the *general procedure B*, **12b** (65 mg, 0.30 mmol), **29** (0.10 g, 0.30 mmol) and sodium triacetoxyborohydride (0.13 g, 0.61 mmol) in dichloroethane (10 mL) afforded the BOM-protected intermediate, which was deprotected with TFA (5.0 mL) in the presence of triethylsilane (5.0 mL) at 73 °C for 4 h to give the **23** (eluent system: 5% MeOH in DCM, 57 mg, 0.14 mmol, 46% yield). ^1^H NMR (400 MHz, DMSO-d_6_) *δ* ppm 1.43–1.56 (m, 1 H, piperdyl-5a-yl), 1.59–1.80 (m, 6 H, 5-CH_3_, piperdyl-4-yl, piperdyl-5b-yl), 2.07 (t, *J* = 12.9 Hz, 1 H, piperdyl-6a-yl), 2.22 (t, *J* = 10.3 Hz, 1 H, piperdyl-2a-yl), 2.56 (t, *J* = 4.6 Hz, 2 H, CH_2_N), 2.68–2.81 (m, 3 H, piperdyl-6b-yl, PhCH_2_), 2.82–2.90 (m, 1 H, piperdyl-2b-yl), 4.31–4.43 (m, 1 H, piperdyl-3-yl), 6.81 (dd, *J* = 8.1, 1.8 Hz, 1 H, Ph), 6.90 (s, 1 H, Ph), 6.95–7.04 (m, 3 H, Ph), 7.12 (t, *J* = 7.6 Hz, 1 H, Ph), 7.28 (t, *J* = 7.8 Hz, 1 H, Ph), 7.38 (t, *J* = 7.9 Hz, 2 H, Ph), 7.68 (s, 1 H, H-6), 11.23 (s, 1 H, NH). ^13^C NMR (101 MHz, DMSO-d_6_) *δ* ppm 12.0 (1 C, 5-CH_3_), 23.9 (1 C, piperdyl-5-yl), 28.1 (1 C, piperdyl-4-yl), 32.3 (1 C, PhCH_2_), 51.0 (1 C, piperdyl-3-yl), 52.2 (1 C, piperdyl-6-yl), 56.5 (1 C, piperdyl-2-yl), 59.2 (1 C, CH_2_N), 108.6 (1 C, C-5), 116.1 (1 C, Ph), 118.5 (2 C, Ph), 119.0 (1 C, Ph), 123.3 (1 C, Ph), 123.9 (1 C, Ph), 129.7 (1 C, Ph), 130.0 (2 C, Ph), 138.0 (1 C, C-6), 142.7 (1 C, Ph), 150.8 (1 C, C-2), 156.5 (1 C, Ph), 156.7 (1 C, Ph), 163.7 (1 C, C-4). HRMS (ESI): *m/z* [M + H]^+^ Calcd. for [C_24_H_27_N_3_O_3_ + H]^+^ 406.2125, found 406.2125.

*5-Methyl-1-(1-(2-(3-phenoxyphenyl)acetyl)piperidin-4-yl)pyrimidine-2,4(1H,3H)-dione* (**26**). To a solution of **10b** (0.2 g, 0.82 mmol) in MeOH (5.0 mL) was added 1M NaOH (5.0 mL), the resulting reaction mixture was stirred at 50 °C for 2 h. After cooling to room temperature, the reaction mixture was treated with 1N aq. HCl to pH 2–3. The generated white precipitate was collected through filtration and dried *in vacuo*. The intermediate (0.17 g, 0.74 mmol), **28** (0.26 g, 0.79 mmol), EDC HCl (0.29 g, 1.5 mmol) and 4-dimethylaminopyridine (DMAP) (9.1 mg, 7.4 μmol) were dissolved in DCM (30 mL), and the reaction mixture was stirred at room temperature for overnight to afford BOM protected intermediate, which was deprotected with Pd/C (10%), H_2_ and HCOOH (0.5%) in i-propanol/H_2_O (10/1 mL) [30] to give **26** (eluent system: 5% MeOH in DCM, 45 mg, 0.11 mmol, 13% yield). ^1^H NMR (300 MHz, DMSO-d_6_) *δ* ppm 1.54–1.83 (m, 7 H, 5-CH_3_, piperdyl-3-yl, piperidyl-5-yl), 2.53–2.67 (m, 1 H, piperidyl-2/6-yl), 3.07 (t, *J* = 10.8 Hz, 1 H, piperidyl-2/6-yl), 3.59–3.85 (m, 2 H, COCH_2_), 4.05 (d, *J* = 13.5 Hz, 1 H, piperidyl-2/6-yl), 4.43–4.56 (m, 2 H, piperidyl-4-yl, piperidyl-2/6-yl), 6.82–6.91 (m, 2 H, Ph), 6.95–7.03 (m, 3 H, Ph), 7.08–7.15 (m, 1 H, Ph), 7.31 (t, *J* = 7.8 Hz, 1 H, Ph), 7.34–7.41 (m, 2 H, Ph), 7.53 (d, *J* = 0.9 Hz, 1 H, H-6), 11.20 (s, 1 H, NH). ^13^C NMR (101 MHz, DMSO-d_6_) *δ* ppm 12.1 (1 C, 5-CH_3_), 29.7 (1 C, piperidyl-3/5-yl), 30.5 (1 C, piperidyl-3/5-yl), 38.9 (1 C, (CO)CH_2_), 40.8 (1 C, piperidyl-2/6-yl), 44.7 (1 C, piperidyl-2/6-yl), 52.1 (1 C, piperidyl-4-yl), 109.1 (1 C, C-5), 116.7 (1 C, Ph), 118.7 (2 C, Ph), 119.5 (1 C, Ph), 123.5 (1 C, Ph), 124.4 (1 C, Ph), 129.9 (1 C, Ph), 130.1 (2 C, Ph), 137.7 (1 C, Ph), 138.2 (1 C, C-6), 150.8 (1 C, C-2), 156.6 (1 C, Ph), 156.7 (1 C, Ph), 163.8 (1 C, C-4), 168.5 (1 C, CO(CH_2_)). HRMS (ESI): *m/*z [M + H]^+^ Calcd. for [C_24_H_25_N_3_O_4_ + H]^+^ 420.1918, found 420.1925.

*1-(3-Phenoxyphenethyl)-4-phenylpiperidine* (**27**). To a solution of **12b** (0.12 g, 0.56 mmol) and 4-phenylpiperidine (0.14 g, 0.87 mmol) in dichloroethane (10 mL) was added sodium triacetoxyborohydride (0.24 g, 1.1 mmol), the resulting mixture was stirred at room temperature for overnight. The reaction mixture was diluted with DCM, and extracted with sat. NaHCO_3_ and brine. The collected organic layer was dried, concentrated and purified to give **27** (58 mg, 0.16 mmol, 29% yield). ^1^H NMR (300 MHz, CDCl_3_) *δ* ppm 1.91 (br. s., 4 H, piperdyl-3-yl, piperidyl-5-yl), 2.12–2.31 (m, 2 H, piperidyl-2a-yl, piperidyl-6a-yl), 2.46–2.62 (m, 1 H, piperidyl-4-yl), 2.65–2.77 (m, 2 H, CH_2_N), 2.83–2.97 (m, 2 H, PhCH_2_), 3.20 (d, *J* = 10.8 Hz, 2 H, piperidyl-2b-yl, piperidyl-6b-yl), 6.82 - 6.95 (m, 2 H, Ph), 6.96–7.07 (m, 3 H, Ph), 7.12 (t, *J* = 7.3 Hz, 1 H, Ph), 7.17–7.46 (m, 8 H, Ph). ^13^C NMR (75 MHz, CDCl_3_) *δ* ppm 33.0 (2 C, piperdyl-3-yl, piperidyl-5-l), 33.2 (1 C, PhCH_2_), 42.4 (1 C, piperidyl-4-yl), 54.1 (2 C, piperidyl-2-yl, piperidyl-6-yl), 60.3 (1 C, CH_2_N), 116.5 (1 C, Ph), 118.8 (1 C, Ph), 119.1 (1 C, Ph), 123.2 (1 C, Ph), 123.6 (1 C, Ph), 126.2 (1 C, Ph), 126.8 (3 C, Ph), 128.4 (2 C, Ph), 129.6 (1 C, Ph), 129.7 (2 C, Ph), 142.1 (1 C, Ph), 145.9 (1 C, Ph), 157.1 (1 C, Ph), 157.8 (1 C, Ph). HRMS (ESI): *m/z* [M + H]^+^ Calcd. for [C_25_H_27_NO + H]^+^ 358.2166, found 358.2164.

## 4. Conclusions

Starting from the earlier reported compound **3**, 19 analogs were synthesized and evaluated for the inhibitory potencies of both *Mtb*TMPK and *M. tuberculosis* (H37Rv) in vitro growths. Selected substituents on the terminal phenyl ring slightly improved the inhibitory potency. Surprisingly, the 3-Cl analog (**21j**) was three-fold more potent than compound **3** as the *Mtb*TMPK inhibitor. Substitution of the distal phenyl ring of **3** and **21n** afforded analogs with superior whole cell antimycobacterial activity compared to **3**. These results provide possible directions for further investigations of *Mtb*TMPK inhibitors as antituberculosis agents.

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
