# Peer review of "1-(1-Arylethylpiperidin-4-yl)thymine Analogs as Antimycobacterial TMPK Inhibitors"

_molecules, 2020, doi:10.3390/molecules25122805_

Round 1
Reviewer 1 Report
Multidrug resistance is a major problem in the treatment of TB and there is a need for novel anti-TB agents which are more effective. Authors synthesized a new series of MtbTMPK inhibitors using a previously reported compound 3 as a starting point. Novel compounds generated were evaluated by enzyme inhibition and in vitro antimycobacterial activity to identify more potent inhibitors of MtbTMPK inhibitors. Docking studies were also performed to predict the specific interactions involved. This study generated new information and can provide new directions for further improvement of MtbTMPK inhibitors.
Overall, the manuscript is very well-written and the data is presented very effectively. Conclusions are well-supported by the data provided.
Author Response
-
Reviewer 2 Report
The manuscript describes the synthesis of 1-(1-Arylethylpiperidin-4-yl)thymine analogs and evaluation for the growth of virulent M. tuberculosis strain (H37Rv) and the capacity of some of them to inhibit MtbTMPK catalytic activity. Modifications of the scaffold of 3 do not afford substantial improvements in MtbTMPK inhibitory activity and antimycobacterial activity. Optimization of the substitution pattern of the D ring of 3 results in compound 21j with improved MtbTMPK inhibitory potencies (3-fold) and H37Rv growth inhibitory activity (2-fold).
The paper is well presented with a comprehensive language and style, the authors properly cite the relevant prior level of the investigations as well as the chemistry is nicely described.
The activity data are promising but low selectivity index - SI (SI values =IC50/MIC) don’t indicate high effectiveness and safety of the new compounds and don't confirm their potential to be used as antibacterial drugs. Anti-mycobacterial agents are considered bacteria selective when the calculated SI is >10 [K. Katsuno, J.N. Burrows, K. Duncan, et al., Hit and lead criteria in drug discovery for infectious diseases of the developing world, Nat Rev Drug Discov. 14 (2015) 751-758.], [M.H. Shaikh, D.D. Subhedar, L. Nawale, et al., 1, 2, 3-Triazole derivatives as antitubercular agents: synthesis, biological evaluation and molecular docking study, Medchemcomm 6 (2015) 1104-1116]. When the compounds showed poor bacterial selectivity as evidenced by their low selectivity indices <10, the results indicate that these analogs kill cells indiscriminately. Thus, the results provide possible directions for further investigation of MtbTMPK inhibitors as anti-tuberculosis agents.
I would recommend the authors present the calculated values of the selectivity index in Table 3 as well as to replace the 7H9 with MICa H37Rv in the second column in the same table.
Additionally, evaluation of the synthesized compounds for their inhibitory potency against Plasmodium falciparum is presented incompletely. As it does not support the above study, this investigation could have been removed from this manuscript.
In the Supplementary part, the HSQC and HMBC NMR spectral DATA are not visible.
In summary, this manuscript is interesting and provides a synthetic foundation for future efforts to developing highly specific anti-tubercular agents. After a major revision, it would be suitable for publication in Molecules.
Author Response
- I would recommend the authors present the calculated values of the selectivity index in Table 3 as well as to replace the 7H9 with MICa H37Rv in the second column in the same table.
Response: We acknowledge the reviewer’s point. The values of the selectivity index are now included in Table 3. We have also replaced “7H9” with MICa (H37Rv) and “MRC-5” with IC50 (MRC-5).
- Additionally, evaluation of the synthesized compounds for their inhibitory potency against Plasmodium falciparum is presented incompletely. As it does not support the above study, this investigation could have been removed from this manuscript.
Response: According to reviewer’s suggestion, the information (line 160 and 161 in original manuscript) on the activity against Plasmodium falciparum has been removed from the manuscript.
- In the Supplementary part, the HSQC and HMBC NMR spectral DATA are not visible.
Response: We apologize for the inconvenience caused by the incompatibility of the HSQC and HMBC NMR spectra in different PDF Reader software. We found the spectra were only visible in Foxit Reader (not visible in Adobe Reader), which may be related to the fact that these spectra were combined and edited in Foxit Phantom PDF. We have now improved the HSQC and HMBC NMR spectra in the supplementary file, which are now visible in both Readers.
Reviewer 3 Report
This work describes design of new derivatives treating previously obtained by Authors 1-(1-arylethylpiperidin-4-yl)thymine as leading structure. Modifications were planned in four parts of lead. Authors have obtained 19 new structures, evaluated their inhibitory potencies for Mycobacterium tuberculosis thymidylate kinase (MtbTMPK) - key enzyme for bacterial survival and Mycobacterium tuberculosis in vitro growth. Selectivity over MRC-5 fibroblasts was evaluated. Docking studies in MtbTMPK of lead and new compounds allowed to analyze interactions of examined compounds with biological target. The subject of this article falls into the scope of Molecules. It is valuable, good written work. My suggestions for minor corrections are:
- Please expand in one place used abbreviations: BOM protecting group, PCC, EDC, DMAP…
- Please comment how were NMR spectra measured of obtained aldehyde building blocks, if samples were purified
- Please add chemical formula for aldehyde building blocks
- Correct abbreviation BOM (instead of Bom) in Scheme 2 for structures 28 and 29
Author Response
- Please expand in one place used abbreviations: BOM protecting group, PCC, EDC, DMAP…
Response: We have defined these abbreviations in parentheses the first time they appear in the manuscript, i.e. benzyloxymethyl (BOM) in line 69, trifluoroacetic acid (TFA) in line 70, pyridinium chlorochromate (PCC) in line 73, dichloromethane (DCM) in line 80, dimethylformamide (DMF) in line 82, N-(3-dimethylaminopropyl)-N′-ethylcarbodiimide hydrochloride (EDC.HCl) in line 92, 4-dimethylaminopyridine (DMAP) in line 93, adenosine triphosphate (ATP) in line 171, thymidine monophosphate (dTMP) in line 172 and nicotinamide adenine dinucleotide (NADH) in line 173.
- Please comment how were NMR spectra measured of obtained aldehyde building blocks, if samples were purified
Response: We didn’t purify the aldehyde building blocks, and thus the NMR spectra of these intermediates are not available. After addition of PCC, the reaction mixture became dark brown to black, indicating that the reaction took place. Through a short silica column, the undesired dark product was removed, and the obtained colorless to yellow oily aldehyde was used without purification.
- Please add chemical formula for aldehyde building blocks.
Response: As requested by the reviewer, we have added the chemical formula for ester/alcohol intermediates and aldehyde building block in the manuscript, section 3.5.
- Correct abbreviation BOM (instead of Bom) in Scheme 2 for structures 28 and 29.
Response: We appreciate the reviewer’s point, and have corrected “Bom” into BOM in Scheme 2 for structures 28 and 29.
Round 2
Reviewer 2 Report
Dear Editor,
The manuscript is well written, the authors properly cite the relevant prior level of the investigations, the chemistry is nicely described, and the Results, Conclusion, and Supplementary sections were updated.
The manuscript is suitable for publication in Molecules after the correction of minor typos.